# Gamma-band synchronization between neurons in the visual cortex is causal for effective information processing and behavior

Eric Drebitz ⬡ ✉, Lukas-Paul Rausch & Andreas K. Kreiter

Successful behavior relies on the brain's ability to process selectively attended information while suppressing irrelevant information. Visual neurons show such functional flexibility by selectively responding to subsets of inputs representing attended objects while ignoring those conveying information about irrelevant objects. Several neuronal mechanisms have been proposed to explain this attention-dependent processing, yet none has been proven as a causal mechanism. One requires precise synchronization between spikes carrying relevant information and the gamma-oscillatory activity in receiving neurons. To investigate its causal relevance, we electrically evoked single volleys of spikes in area V2 of two male macaque monkeys performing a selective-attention task and recorded neuronal activity in downstream area V4. Strongly depending on the γ-phase, when these additional spikes arrived in V4, they impaired monkeys' performance and evoked a spiking response. This establishes the causal relevance of subtle changes in spike timing, specifically by phase synchronization, for neuronal mechanisms serving cognitive processes.

Small receptive fields (RFs) and precise retinotopy in the early visual cortex enable the parallel processing of multiple closely spaced stimuli in separate neuronal populations (Fig. 1A). However, the ubiquitous convergence of feedforward connections along cortical processing pathways results in increasingly larger RFs for downstream neurons[1]. These convergent connections enable information integration across larger visual field regions; however, they also often convey signals representing different objects that compete to be processed by the same neurons[2,3].

In the visual cortex, selective attention resolves this competition in favor of the most behaviorally relevant stimulus: Essentially, neurons with multiple stimuli in their RFs respond as if only the attended stimulus is present[4-7] by selectively processing the signals obtained by that subset of their afferent inputs, which provides information on the attended stimulus while suppressing other signals[8-10].

Several mechanisms have been proposed to explain the remarkable functional flexibility of cortical processing[11-15]. One mechanism suggests that preferential routing and processing of the selected afferent signals result from precise timing of the corresponding spikes relative to the oscillatory cycles of the receiving neurons' γ-band activity (30–90 Hz)[11,16-18]. The rationale is that local γ-oscillatory activity results in a rhythmic modulation of neuronal excitability. Thus, afferent spikes should be most effective when they arrive during the sensitive phase of the excitability cycle of the receiving neuron. Accordingly, inputs not arriving during this phase should have considerably less influence on the receiving neuron[11,19,20]. However, other proposed mechanisms do not rely on precise temporal alignment between afferent inputs and receiver neurons[12,21], and the functional relevance of γ-band synchronization is under debate[22-24].

Cognitive Neurophysiology, Brain Research Institute, University of Bremen, Bremen, Germany. ✉e-mail: drebitz@brain.uni-bremen.de

**Fig. 1 | Schematic of intracortical microstimulation (ICM), intracortical recordings, and the behavioral task paradigm. A** Connection scheme of converging input from V2 to retinotopically matching V4 neurons. The two stimuli below activate separate V2 populations. A micro-electrode indicates the V4 recording site. The micro-electrode with a lightning symbol illustrates the V2 site receiving the ICM. The ICM-evoked spikes originating in V2, shown on the right in yellow, arrive at the recorded V4 site at random V4 γ-phases (black traces, right). **B** Visual stimulation comprises two stimuli within the V4 RFs, which continuously change their shape during a trial. Each stimulus could become the target of attention. In this example, attention was cued to the stimulus represented by V2 neurons that did not receive ICM. The animals had to hold fixation on the central fixation point (FP). The illustrations of the focus of attention and the V4 and V2 RF-outlines are shown here for illustration, but do not appear on the screen. **C** Morphing sequence of a target stimulus. Before trial onset, a spatial cue indicated the location of the target stimulus in the upcoming trial. The animals initiated the trial by fixating on the central FP and pressing a lever. The cue then disappeared, and the baseline period (Base) began. After the end of the baseline period, the stimuli appeared on the screen with their initial shape and were presented statically

during the 'Static' interval. Subsequently, the stimuli morphed continuously into other shapes for up to four MC (MC 1–4). When the initial shape of the target stimulus reappeared, the monkeys had to release the lever during a 510-ms response time window (red rectangle) to receive a reward. In 2/3 of the trials, a single, biphasic, cathodic first ICM pulse of ±15 μA or ±25 μA was applied at a random time within the second half of each morph cycle between 50 ms and 450 ms before the end of the MC (highlighted in yellow). **D** Illustration of the retinotopic alignment between RFs of the V4 and V2 recoding sites, based on the average normalized spiking responses to two stimuli shown in separate trials. Both stimuli were located within the V4 RFs, with one also falling within the V2 RF, as shown in the insets in the upper right corner of both panels. The traces show the normalized entire spiking activity (ESA) pooled across animals for all V4 (left panel) and V2 recording sites (right panel). For V4 sites ($n = 146$, $p < 0.05$), the traces show significant responses to both stimuli during the morphing cycles 2 and 3, while for V2 sites ($n = 20$, $p < 0.05$), they show significant responses to one stimulus. Dashed traces correspond to the stimulus inside V2 RF, and solid traces to the other stimulus, as indicated by the insets. The gray shading indicates ±1 SEM.

Previous theoretical studies have demonstrated the plausibility of a γ-phase-dependent input gain modulating mechanism for selecting among competing afferent streams of information[25–30]. In line, local field potential (LFP)-based studies have confirmed an attention-

dependent enhancement of interareal γ-band coherence between receiver populations and selected subsets of their inputs[9,10,16]. Furthermore, intra- and interareal γ-band coherence parameters, like phase and amplitude, correlate with faster response times (RTs) and

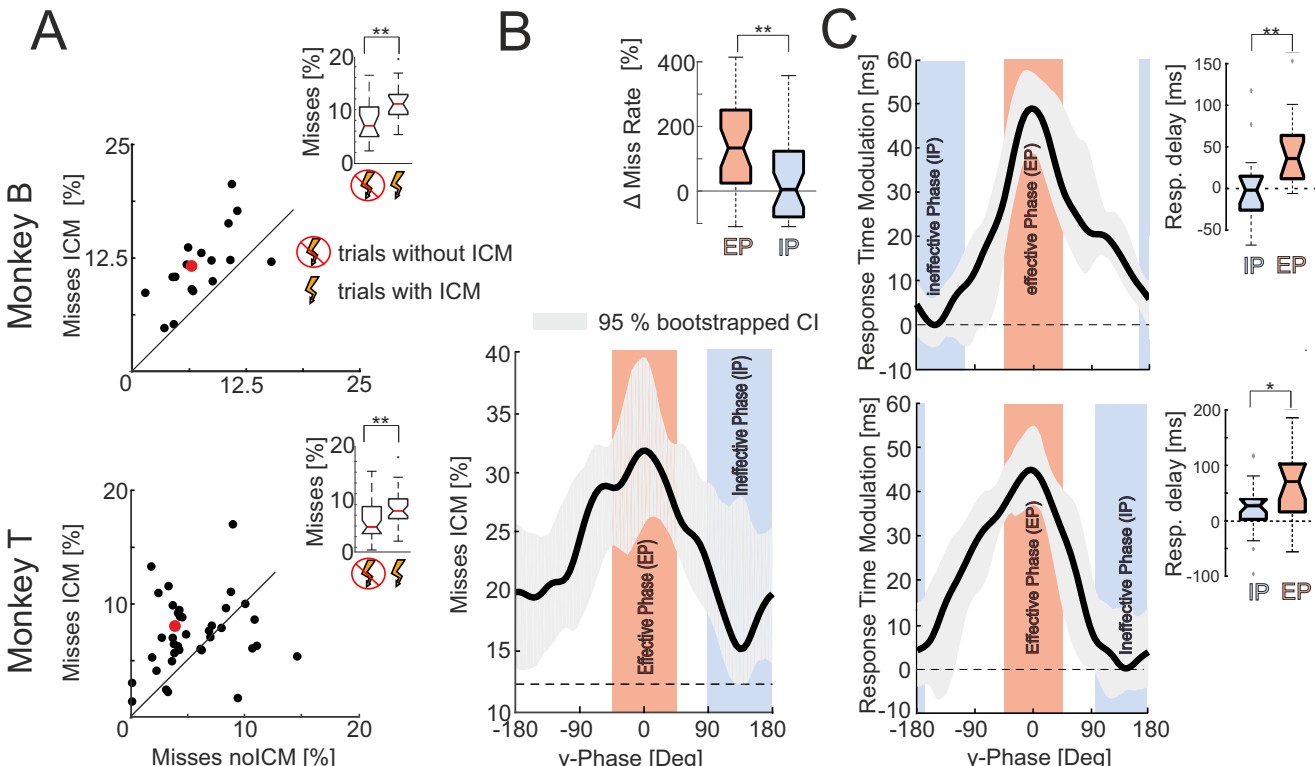

**Fig. 2 | Effect of ICM on the proportion of misses and RTs. A** The scatterplots show the proportion of misses in trials with and without ICM for the recording sessions of both monkeys (monkey B: $n = 17$, monkey T: $n = 40$). The red dots indicate the median proportions. The insets show the distribution of the proportion of misses for both ICM conditions. The uncrossed and crossed lightning symbols indicate the results from trials with and without ICM, respectively (monkey B: $p = 0.0030901$, $z = 2.9586$; monkey T: $p = 0.011505$, $z = 2.527$, both Wilcoxon signed-rank tests, two-sided). **B** γ-phase-dependence of ICM's efficacy to cause misses (data pooled across both animals). The graph shows the median percentage of trials resulting in misses of all trials that resulted in correct responses or misses as a function of the V4 populations' γ-phase after the ICM pulse across recording sessions. The shaded area represents the 95% confidence interval, estimated via bootstrapping (1000 resamples with replacement). We centered the median curve's maximum at 0° for better visibility of the unimodal shape. The upper right inset shows the proportion by which the number of misses differed in ICM trials that belong to the effective (EP, red) and ineffective (IP, blue) phase range from the expected number observed for trials without ICM ($n = 36$ recording sessions,

$p = 0.0017691$, $z = 3.1265$, Wilcoxon signed-rank test, two-sided). **C** γ-phase dependence of ICM's efficacy in causing RT delays. The graphs show the median RTs across recording sites (upper = monkey B; lower = monkey T), plotted as a function of the V4 γ-phase after the ICM pulse. The shaded area represents the 95% confidence interval, estimated via bootstrapping (1000 resamples with replacement). We centered the median curves' maxima at 0° for better visibility of the unimodal shapes. To depict the modulation depth of the RTs across phases, the minimum of the curve is set to zero. The insets show the RT delays across recording sessions with respect to the median RTs in trials without ICM for trials with ICM belonging to the effective (red) and ineffective (blue) phase range (monkey B: $n = 16$, $p = 0.0021483$, $t = 3.3579$; monkey T: $n = 18$, $p = 0.011919$, $t = 2.6571$, paired Student's $t$-tests, two-sided). The boxplots (**A**–**C**) show the medians and interquartile ranges (IQR); whiskers extend to data points within 1.5× IQR from the quartiles, and data points beyond this range are considered outliers. **B**, **C** The ±45° wide phase ranges centered around the curves' maxima and minima define the effective (red) and ineffective (blue) phase ranges, respectively. **Indicates significance at $p < 0.01$ and *$p < 0.05$. Source data are provided as a Source Data file.

firing rate enhancements[31–33]. However, the crucial question of whether the γ-phase at which spikes arrive at postsynaptic neurons determines their effectiveness in signal transmission and ultimately on behavior remains open[34–36].

To directly test for such a causal role, we investigated whether the ability of an additional short input signal to disturb cortical information processing depends on the current phase of the receiving neurons' γ-oscillatory excitability cycle. For this purpose, single electric intracortical microstimulation (ICM) pulses were applied in the upper layers of visual area V2 in two monkeys to evoke brief volleys of synchronous spikes[37–39] while the monkeys performed an attention-demanding shape-tracking task[16,40]. These spikes provide an externally evoked, transient, and potentially disturbing input signal to the downstream area V4 that is not caused by internal neural processes (Fig. 1A, indicated by yellow-colored spikes). At the same time, we recorded neural activity at V4 sites that retinotopically matched the microstimulated V2 location, as illustrated by the pattern of spiking responses for V4 and V2 sites in Fig. 1D.

If the γ-phase of these upstream V4 neurons plays a causal role in determining the efficacy of an incoming signal, then the negative impact of the artificially evoked spikes in V2 on the information processing of these neurons and subsequent behavioral deterioration should depend on this phase (Fig. 1A). The shape-tracking task (Fig. 1B, C) required that the monkeys observe one of four stimuli (1° diameter each), which continuously morphed between different complex shapes (Fig. 1C). The animals had to respond to the reappearance of the initial shape of the target stimulus after two to four morphing cycles (MCs, Fig. 1C) by releasing a lever. Two of the four stimuli were located within the RF of the recorded V4 neurons, which responded to each of them when shown individually (Fig. 1D, left panel). Consequently, the two stimuli competed for processing by these V4 neurons, thereby stimulating mechanisms to select relevant inputs while suppressing irrelevant inputs. One of these stimuli also covered the smaller RF of the ICM site in V2 (Fig. 1A, B). In the ICM trials, a single biphasic current pulse (±15 or ±25 μA) was randomly delivered between 50 ms and 450 ms before the end of each MC (Fig. 1C, highlighted yellow).

## Results

### Perceptual interference evoked by ICM is γ-phase-dependent

The weak, single ICM pulses were sufficient to increase the small proportion of trials in which monkeys failed to respond (misses) by more than 60% for both monkeys (Fig. 2A; the median proportion of misses increased by 64.2% and 60.5%, respectively; monkey B: $n = 17$ sessions, median noICM: 6.7%, median ICM: 11.0%, $p = 0.0030901$, $z = 2.9586$; monkey T: $n = 40$ sessions, median noICM: 4.3%, median ICM: 6.9%, $p = 0.011505$, $z = 2.527$; both Wilcoxon signed-rank tests, two-sided). The ICM pulses did not significantly affect the number of fixation failures, while the percentage of false alarms decreased significantly for monkey T but not for monkey B (Supplementary Table 1). These findings indicate that the volley of spikes evoked by a single ICM pulse can effectively interfere with information processing and, ultimately, with behavior.

Importantly, for the sessions in which ICM caused increasing numbers of misses, the effectiveness of ICM-evoked spikes in interfering with stimulus processing and behavior depended strongly on the phase of the γ-oscillatory excitability cycle of the receiving V4 neurons (Fig. 2B). The difference in the number of misses between the most effective and the ineffective phase ranges was significantly larger than expected by chance ($n_{shuf} = 10{,}000$, $p = 0.0021$, bootstrap hypothesis testing, one-sided). For the most effective phase range of the excitability cycle (highlighted in red in Fig. 2B), the proportion of misses (31.9%) was more than twice as high as for the least effective phase range (highlighted in blue, 15%). Moreover, for the least effective phase range, the proportion of misses barely exceeded the level observed in trials without ICM (12.5%; vertical dashed line in Fig. 2B), and the difference did not reach significance (median: 3.2%, $n = 36$, $p = 0.18428$, $z = 1.3277$, Wilcoxon signed-rank test, two-sided; Fig. 2B, upper panel). Conversely, ICM-evoked spikes arriving in the most effective phase range significantly increased the median number of misses by 133% ($n = 36$, $p < 0.001$, $z = 4.7453$, Wilcoxon signed-rank test, two-sided). The effectiveness of ICM in causing misses decreased significantly by approximately 98% during the ineffective phase range compared to that during the effective phase range ($n = 36$, $p = 0.0017691$, $z = 3.1265$, Wilcoxon signed-rank test, two-sided).

The γ-phase-dependent effect of ICM on behavior did not affect only performance. Consistent with the γ-phase dependence of misses, we found a γ-phase-dependent increase in RTs for both animals (Fig. 2C). For both animals, the modulation of RTs between effective and ineffective phase ranges was significantly larger than expected by chance (upper panel monkey B: $n_{shuf} = 10{,}000$, $p = 0.0004$, lower panel monkey T: $n_{shuf} = 10{,}000$, $p = 0.0005$, both bootstrap hypothesis testing, one-sided). ICM-evoked spikes arriving during the effective phase range caused RT delays of 45–70 ms (Fig. 2C; upper panel monkey B: 44.6 ms, $n = 16$, $p = 0.00031362$, $t = 4.072$; lower panel monkey T: 69.9 ms, $n = 19$, $p = 2.0879 \times 10^{-5}$, $t = 4.892$, paired Student's $t$-tests, two-sided), while those arriving in the ineffective phase range did not significantly affect RTs (monkey B: −0.4 ms, $n = 16$, $p = 0.97113$, $t = 0.036489$; monkey T: 21 ms, $n = 19$, $p = 0.063463$ ms, $t = 1.915$; paired Student's $t$-tests, two-sided). In addition, the differences between RTs for both phase ranges were highly significant (monkey B: 45.0 ms, $n = 16$, $p = 0.0021483$, $t = 3.3579$; monkey T: 48.9 ms, $n = 18$, $p = 0.011919$, $t = 2.6571$; paired Student's $t$-tests, two-sided) and this effect was equally strong for RTs following the reappearance of the initial shape at the end of MCs 2 and 3 (Supplementary Fig. 5).

To examine if such substantial effects of single ICM pulses on RTs might be linked to ICM-induced changes in the direction of gaze, we compared gaze shifts following ICM application with those during periods without ICM application (see Supplementary Fig. 1). For both animals, there were neither phase-dependent nor phase-independent systematic effects of the ICM pulses on gaze positions, ruling out the possibility that the observed ICM-induced effects on RTs are related to ICM-evoked alterations of gaze positions.

We also tested whether the γ-phase-dependent effect of ICM on RTs could be attributed to specific frequencies within a broad γ-frequency range. We observed similar effect sizes for frequencies between 40 Hz and 80 Hz, while the effect vanished toward the higher γ- and β-frequency ranges (Supplementary Fig. 8).

The pronounced dependence of ICM-evoked behavioral effects on the phase of γ-oscillatory activity in V4 suggests that the critical interference between the ICM-evoked spike volleys and behaviorally relevant neuronal stimulus processing occurred in the local neuronal population around the V4 recording site. The alternative hypothesis that the critical interference occurred directly within the stimulated V2 population and depended on the V2 γ-phase is not plausible. When both the V4 and afferent V2 population process a target stimulus, the known phase-coupling between the receiving V4 populations and relevant afferent input[9,16] results in V4 γ-phases that mirror the allegedly relevant γ-phases of the V2 population. Conversely, when the afferent population processes a distractor, there is essentially no interareal phase coupling and, hence, no relation between the V4 and V2 γ-phases. Consequently, the observation of a similarly strong dependence of behavioral ICM effects on the V4 γ-phase, even when the stimulated V2 population processes a distractor, contradicts that the critical interference occurred in V2 (see Supplementary Fig. 3 and Supplementary Note 2).

Finally, we excluded the hypothetical possibility that the timing of ICM pulses, combined with the time course of the visual stimulation and the potentially induced γ-oscillatory activity, may have resulted in the observed γ-phase-dependent RT modulation. We randomly assigned ICM pulse times to trials without ICM and assessed potential γ-phase-dependent RT modulations, similar to the evaluation of the ICM data. We did not observe significant RT modulation for trials without ICM for either animal (Supplementary Fig. 4 and Supplementary Note 3).

### Effect of stimulation on spiking activity and the LFP

The strongly phase-dependent impact of single ICM pulses on behavior suggests that the evoked spike volley's efficacy for disturbing the V4 neurons' ongoing processing of the relevant stimulus depends on the phase of their γ-oscillatory excitability cycle when the synaptic input arrives in V4. To investigate whether and how V4 neuronal activity is affected, we calculated the average ICM pulse-evoked deviation of the spiking activity from its undisturbed time course and assessed how such deviations depend on the different phases of the γ-cycle. These γ-phases were calculated for the time 10 ms after the ICM pulse, which is close to the time when the ICM-evoked spikes arrived in V4 (see "Methods" for details). The resulting heatmap in Fig. 3A shows that V4 neurons respond shortly, between 10 ms and 15 ms, after applying the ICM pulse in V2. Importantly, a substantial response occurred only when the phase of the V4 neurons' excitability cycle was within a limited range, approximately 90° wide (from 75.2° before to 14.8° after the cycle's peak). The average entire spiking activity (ESA[6,41,42]) response in the 2 ms interval centered around its maximum at 12.6 ms (mean $z$-score = 3.83; Fig. 3A, right panel) was significantly larger than for the corresponding maxima taken across all phases of the corresponding ESA from trials without ICM (bootstrapped mean $z$-score = $1.5 \pm 0.64$ SD, $n_{shuf} = 15{,}000$, $p = 0.0015$, bootstrap hypothesis testing, one-sided). The ICM-evoked peak occurred when the V4 population's excitability cycle was 15° before its maximum, 10 ms after the ICM pulse. Outside this most effective phase, the enhanced firing quickly decayed and finally disappeared during large parts of the γ-cycle (Fig. 3A, right panel). Accordingly, the phase dependence of ICM-evoked response was far beyond confidence boundaries obtained from trials without ICM (normalized sum vector (nSV) ICM data: 0.58, median nSV non-ICM data: 0.0114, 95% CI: 0.0111−0.0118, $n_{shuf} = 15{,}000$, $p < 6.67 \times 10^{-5}$, bootstrap hypothesis testing, one-sided).

In line with the additional currents expected as a consequence of the additional spikes evoked in V4, there was a significant response in

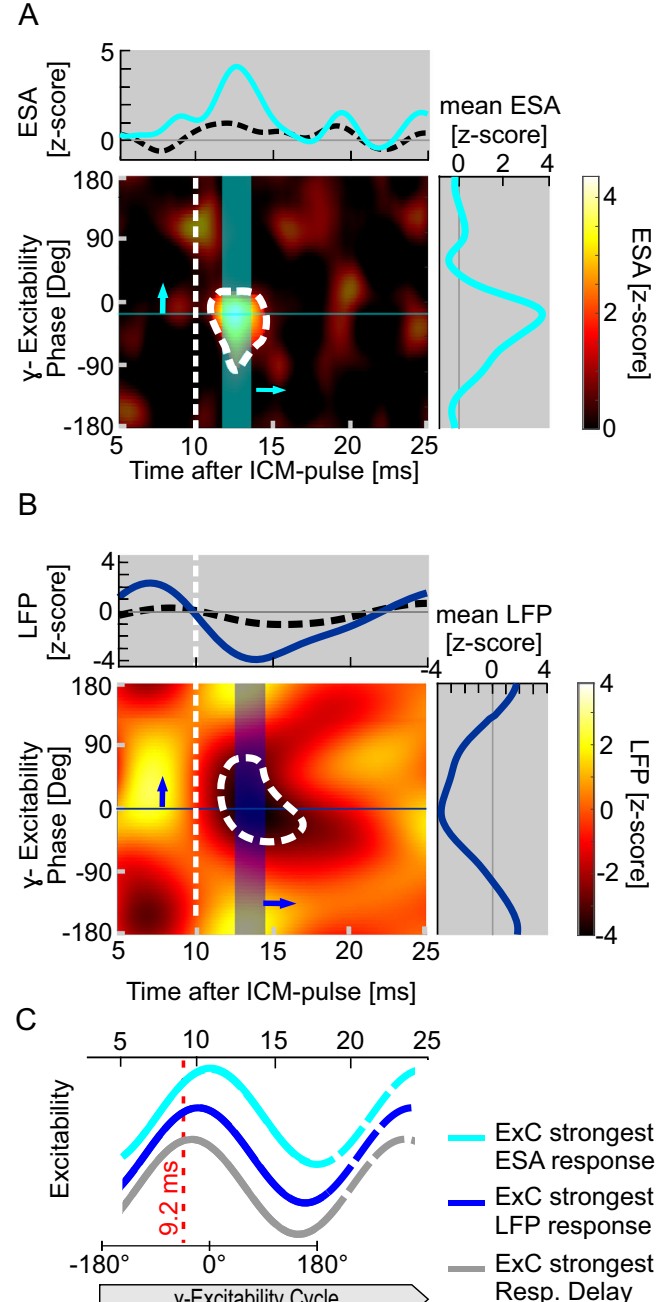

**Fig. 3 | γ-Phase-dependent ESA and LFP response of V4 neurons to ICM pulses.**

**A** The heatmap shows the time course of the ICM-evoked ESA exceeding the corresponding unperturbed ESA depending on the phase of the γ-oscillatory excitability cycle at 10 ms (vertical dashed line) after the ICM pulse. The x-axis indicates the time after the ICM pulse, and the y-axis the γ-phase at 10 ms. The average z-transformed ESA values are color-coded (only positive values are shown for clarity). The only significant response is a short peak (highlighted and encircled by a dashed line; $p = 0.0017$, $n = 10,000$, bootstrap hypothesis testing, one-sided) that only occurs when the phase of the excitability cycle immediately preceding the response lies within a narrow phase range. Top panel: The blue curve illustrates the time course of the strongest average response observed at an excitability phase of −15°, as indicated in the heatmap by the horizontal thin blue line crossing the peak. The comparatively flat, black dashed line represents the average time course after the ICM pulse across all ESA responses, regardless of the relation between the ICM pulse and the phase of the γ-oscillatory excitability cycle 10 ms later. Right panel: the dependence of response strength on the excitability phase, taken as the mean value across a 2 ms window centered at the peak response (highlighted by the vertical blue area in the heatmap). **B** Same as in (**A**), but for the LFP. The heatmap shows the deviation of the LFP in trials following ICMs and the LFP of trials without ICMs, but taken from the same times as a function of the phase of the γ-oscillatory excitability cycle at 10 ms. The area with a significant difference is encircled by a dashed line ($p = 0.0005$, $n = 10.000$, bootstrap hypothesis testing, one-sided). Top panel: the dark blue curve shows the time course of the strongest z-scored LFP response at the γ-phase of 0° at 10 ms as indicated by the horizontal dark blue line. Right panel: same as in (**A**) but for the average z-scored LFP response. All other conventions are equal to (**A**). **C** Schematic illustration of the phase progression of the V4 population's γ-oscillatory excitability cycle (ExC) associated with the strongest ESA response (light blue line), the strongest LFP response (dark blue line), and the strongest RT delay (dark gray line). The red dashed line indicates the excitability phases during which the synaptic input caused by the ICM-evoked V2 spikes is expected to arrive. During these phases, near the excitability peak and shortly before the ESA response onset, the synaptic input causes the maximal effects. Source data are provided as a Source Data file.

the LFP (Fig. 3B, encircled by a dashed line) during approximately the same time and for similar γ-phases as observed for the spiking response. The strongest deviation of the LFP following ICM pulses occurred at 13.8 ms after the ICM pulse (z-score = −4.036), reflecting a significantly more negative average extracellular potential (12.8–14.8 ms, mean z-score: −3.95) compared to the LFP from corresponding trials without ICM (mean z-score: −1.32 ± 0.72 SD, $n_{shuf} = 10,000$; $p = 0.0005$, bootstrap hypothesis testing, one-sided). The strongest negative deflection of the LFP occurred when the V4 population's excitability cycle was at its peak (at 10 ms after the ICM-pulse). While we observed this significant ICM-pulse-evoked, γ-phase-dependent increase in spiking activity and the corresponding LFP, we found no evidence of a substantial effect of ICM on the phase progression of γ-oscillatory activity itself (Supplementary Fig. 6 and Supplementary Note 4).

To determine the γ-phase of the excitability cycle when excitatory postsynaptic potentials (EPSPs) are most effective in triggering

additional neuronal activity and disrupting behavior, we estimated the latency between ICM application and EPSP generation in V4. The strongest ESA response showed an onset latency of 10.7 ms (Fig. 3A, top panel, blue curve). Assuming that the somatic summation of the EPSPs and the generation of the initial spikes require approximately 1.5 ms[43,44], the EPSPs caused by the ICM-evoked spike volley arrived with a latency of about 9.2 ms. Consequently, the phase of the γ-oscillatory excitability cycle in which incoming EPSPs are most effective in evoking additional spikes is approximately 36° before its peak (Fig. 3C, red vertical dashed line). The corresponding phase associated with the maximum effect on LFPs was 21° before peak excitability. Moreover, the phase associated with the strongest prolongation of RTs taken at the same latency for the same sessions as used for spiking activity and the LFP was 15° before the peak, and for the corresponding maximum of misses, 1° after the peak.

The finding that ICM-evoked input arriving near the peak of the V4 neurons' excitability cycle most effectively perturbs the neuronal spiking activity in V4 and the monkeys' behavior raises the question of whether the additional spiking activity directly contributes to the observed behavioral impairment. In support of such a relationship, we found that the delay of RTs caused by ICM-evoked input during the effective phase (−15° ± 45° at 9.2 ms, see Fig. 3A) depended on the additional spiking activity caused by this input within the response window (9.2–15.2 ms after the ICM pulse; Fig. 4). In trials where such ICM pulses caused more than the average number of spikes observed in trials without ICM, the median RT increased by 55.98 ms compared to the median RT of trials without ICM but also an above average number of spikes in the response window (Fig. 4, left box plot; $n = 36$, $p = 0.00053676$, $z = 3.4617$, Wilcoxon rank sum test, two-sided). In contrast, in trials where these ICM-pulses did not cause an increase in the number of spikes (indicating an unsuccessful interference with stimulus processing despite the same timing of ICM pulses to the

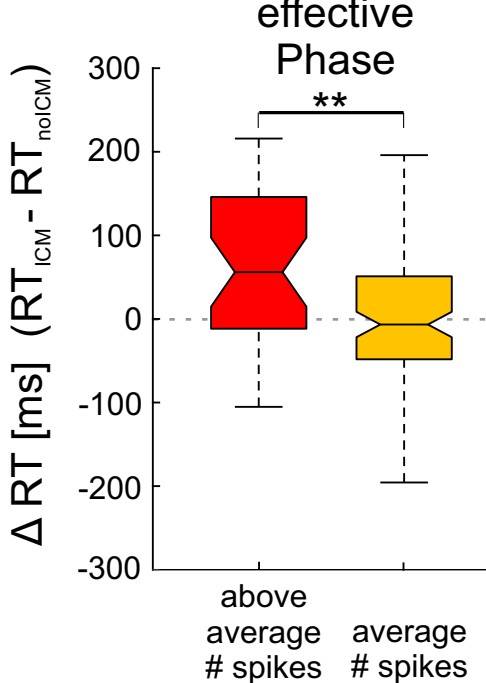

**Fig. 4 | Dependence of RT-delay on ICM-evoked spikes in V4.** The two groups represent the differences in RTs between trials with and without ICM application. The red box represents the difference between RTs of trials, in which the ICM-evoked V2-spikes arrived within the effective V4 γ-phase ($-15° \pm 45°$) and resulted in an above average number of spikes within the following 6 ms (9.2–15.2 ms) and those trials without ICM application, but with naturally occurring above average number of spikes within the same time period and following the same γ-phase. The orange box shows the same differences, but for trials and associated RTs in which the ICM-evoked synaptic input did not cause above average spiking activity in V4 and corresponding trials without ICMs that exhibited an average number of spikes ($n_{above\ average} = 36$, $n_{average} = 105$, $p = 0.002201$, $z = 3.0617$, Wilcoxon rank sum test, two-sided). The boxplot shows the medians and IQR; whiskers extend to data points within 1.5× IQR from the quartiles. $^{**}$Indicate significance at $p < 0.01$. Source data are provided as a Source Data file.

effective phase), the median RTs did not change significantly (Fig. 4, right box plot; median $-6.66$ ms, $n = 105$, $p = 0.25178$, $z = 1.146$, Wilcoxon rank sum test, two-sided). The RT difference between the two trial groups was highly significant, with a median difference of 62.6 ms ($p = 0.002201$, $z = 3.0617$, Wilcoxon rank sum test, two-sided). Importantly, for both monkeys, the successfully increased median ICM-evoked spiking activity did not differ significantly from the above-average median spiking activity in the non-ICM trials used for comparing RTs (differences in spiking activity monkey B = 6.6 %, $p = 0.146$, monkey T = 8.1% $p = 0.129$, $n_{shuf} = 1000$, bootstrap hypothesis testing, one-sided). This indicates that the behavioral impairment results from the unpredictable perturbation of the dynamics of cortical processing caused by the ICM-evoked input arriving during the effective phase of the V4 neurons rather than from the higher spike rate evoked by this interference.

## Discussion

The current study demonstrates a causal relation between the phase of γ-oscillatory activity in a local population of cortical neurons when synaptic input arrives and the strength of its effect on neuronal activity and perceptual behavior. Single volleys of spikes, evoked in supragranular area V2 by single electric ICM pulses, briefly increased the spiking activity within the postsynaptic local populations of V4 neurons, caused corresponding LFP responses, and resulted in longer RTs and a higher probability that monkeys failed to detect the

reappearance of the relevant shape. These effects were most pronounced if the additional input arrived around the peak of the V4 neurons' γ-oscillatory excitability cycle. The farther γ-phases were from the peak, the more the efficacy of ICM evoked spikes declined until essentially no additional spiking activity and prolongation of RTs were caused by the ICM-evoked input. This shows that in cortical networks processing behaviorally relevant stimuli during a demanding perceptual task, the instantaneous phase of local γ-oscillatory activity strongly modulates the efficacy of afferent synaptic transmission.

The behavioral impairments following an ICM pulse likely result from the perturbation of the activity pattern in the local population of V4 neurons. The consistent dependence of both effects on the γ-phase of the local V4 population strongly suggests this direct link. Moreover, the finding that the extent of RT-prolongation following ICM-evoked synaptic input during the effective γ-phase depends on the magnitude of the resulting additional activity in the V4 neurons provides compelling evidence for this relationship.

Several other ways of causing the γ-phase-dependent behavioral impairments are not supported by the results: (1) the ICM-evoked additional spikes did not sufficiently alter activity patterns in area V2 to cause the observed behavioral impairments. If this were the case, ICM applied to the V2-population processing the distracter stimulus should have resulted in much less behavioral impairment than ICM applied to the V2-population processing the target stimulus. Contrary to this expectation, no significant difference was observed between these conditions. Furthermore, the single, low-amplitude ICM pulses applied in V2 likely triggered spikes in only a small number of axons[45], far fewer than the number of neurons within a V2 column[46]. (2) For similar reasons, ICM-induced phosphenes that might impair the perception of a stimulus's shape are unlikely to explain the behavioral impairments. They are typically induced by pulse trains extending over several hundred milliseconds, with frequencies between 100 Hz and 300 Hz, and not by single pulses[47–49]. Moreover, impairments by phosphenes should not be independent of their location close to the target or the distracter stimulus. (3) The lack of ICM-induced changes in gaze position rules out the possibility that ICM-induced eye movements explain the observed behavioral effects.

Thus, the brief activity changes during stimulus processing in V4 most likely account for the γ-phase-dependent delays in RTs and the increased rate of missed responses, as the altered activity patterns in V4 momentarily deviate from those associated with normal shape processing. Additionally, the similarity of the γ-phase-dependent prolongation of RTs due to ICM across different MCs suggests that the additional input during the effective phase interferes with the fundamental neuronal processing of stimulus information, regardless of factors such as attentional expectancy, which changes along the trial.

Whether the ICM-evoked activity modulations in V4 reflect meaningful alterations of shape representations that encode different shapes or correspond to disorganized states that require some additional time to settle again into meaningful states encoding the current stimulus shape remains an open question. The small, essentially random set of afferent axons, each contributing a single additional spike to the local V4 network, the short duration of the activity modulation of only ~5 ms, and the absence of a substantial and consistent increase in false alarms collectively suggest that a meaningful alteration of the perceived shape is unlikely.

The study's behavioral and physiological results demonstrate that the precise temporal structures of neuronal activity and their subtle differences in complex cortical networks are not functionally irrelevant noise or epiphenomenal side effects of neural processing. Instead, the causal consequences of subtly altering these finely tuned timing relationships enable spatiotemporal activity patterns in neuronal networks to selectively determine and modify the effective strength of connections within these networks, thereby influencing their signal and information processing capabilities. Indeed, numerous studies

have documented systematic changes in the spatiotemporal structure of neuronal activity, including selective synchronization and desynchronization of oscillatory activity, that imply meaningful changes in information processing during cognitive processes such as perception or selective attention[9,10,16,32,33,50,51]. Accordingly, γ-phase-dependent modulation of neuronal interactions appears to be a widely employed basic mechanism supporting flexible signal routing, dynamic information processing, and adaptive network configuration[6,52–55].

Furthermore, the causal effect of the γ-phase on the efficacy of afferent inputs implies that the input gains of neurons can be modulated not only by modulating the afferents' firing rates but also by changing the γ-phase of the postsynaptic neurons during which the afferent input arrives. Our results show that depending on how much the input arrival deviates from the optimal γ-phase, the resulting effects decrease and largely vanish. Thus, changing the arrival time by a few milliseconds with respect to the postsynaptic γ-phase enables powerful modulation of the receiver neuron's input gain. Numerous neuronal information processing models assume such modulations of the input gain[26,30,56–59]. For example, the important and widely used divisive normalization mechanism[13,21] often requires strongly different and flexibly changing the weights of the various input signals to implement, for instance, attention-dependent selective processing of one out of several afferent signals[13,60,61]. The γ-phase-controlled input gain observed here flexibly supports the often-required large gain changes without requiring modulations of the afferent inputs' firing rates, which would be implausibly large. The functional role of the phase at which the input arrives is further supported by studies entraining β-oscillations in the motor cortex with trains of phase-locked stimulation pulses, showing subsequent short-term synaptic plasticity[62], changes in β-power, and subsequent β-power correlated behavior[63] depending on the stimulated β-phase. The γ-phase dependence of information routing and processing has also practical consequences for future bidirectional brain-computer interfaces. Feeding information into a working brain, as a visual prosthesis would do, requires that the afferent signals arrive during the effective phases of the targeted neurons' γ-oscillatory cycles to ensure effective transmission and joint processing with the target set of brain signals.

## Methods

### Surgical preparation and ethical statement
Two male macaque monkeys (*Macaca mulatta*) received a head-post, and a recording chamber was implanted above visual areas V4 and V2 under aseptic conditions. The target areas were identified based on anatomical MRI scans. Monkey T, aged 14, was housed in an indoor facility with visual and auditory contact with other monkeys in the same compartment. Monkey B, aged 15, was co-housed with another monkey for direct social interaction within the same facility. The facilities provided various enrichments to promote the physical and psychological well-being of the monkeys. These enrichments included various climbing opportunities, as well as access to toys and games for cognitive stimulation and to encourage natural behaviors.

All procedures complied with the German Animal Welfare Act (TierSchG) and the European Directive 2010/63/EU and were approved by the local authorities (Senator für Gesundheit, Bremen, Germany).

### Visual task paradigm
The animals performed a highly demanding shape-tracking task (Fig. 1B, C). A spatial cue (annulus with a diameter of 1°, 0.04°-line width, Fig. 1C) indicated the location of the behaviorally relevant stimulus in the upcoming trial. The cue and a central FP (FP, 0.15° × 0.15°, 2.45 Cd/m²) were presented on a gray background with 0.03 Cd/m² (Fig. 1B). The animals had to fixate on the FP and press a lever to start the trial, upon which the cue vanished. After a baseline period of 1050 ms with no stimulus on the monitor but the FP, three or four

stimuli, each with a complex-shaped contour (-1.0° diameter, line width 0.25°, 3.8 cd/m², Fig. 1B, C), appeared at isoeccentric locations on the screen. Correspondingly, either one or two stimuli were placed at two locations within the same V4 RF, one of which was also within the RF of the V2 neurons at the stimulation electrode. After the static presentation, the initial shapes of the stimuli were altered continuously to different shapes within 1 s morph cycles (MC). Each trial included two to four MCs, and the animals were rewarded if they responded to the reappearance of the initial shape of the cued stimulus by releasing the lever (Fig. 1C). For each trial, the shapes were randomly selected from a set of 8 or 13 shapes for monkey T and monkey B, respectively. For monkey T, all shapes of the set could become the target shape, while for monkey B, two could become the target shape. The likelihood of a location getting cued was equal across locations. For a more detailed task description, see Drebitz et al.[6], and for a detailed description of the morphing process, see Taylor et al.[40]

Single ICM pulses were applied in two-thirds of the trials. The trials were terminated without reward if the animals released the lever before (false alarm) or after (misses) the response window or moved their direction of gaze by more than 0.5° from the FP (fixation error).

### Experimental setup and recording procedure
The animals sat in custom-made primate chairs that were placed 95 cm (monkey B: 93 cm) in front of a 20-inch CRT monitor (1280 × 1024 pixels, 100 Hz). Video-oculography was used to monitor the eye positions (IScan Inc., Woburn, MA, USA). Up to four epoxy-insulated tungsten microelectrodes (125 μm diameter, 1–3 MΩ, FHC Inc., Bowdoin, ME, USA) were used for acute recordings in V4. A single glass-insulated tungsten microelectrode (125 μm diameter, 1 MΩ, FHC Inc., Bowdoin, ME, USA) was used for ICM application in V2. It was held by a custom-made, adjustable microelectrode drive for semi-chronic recordings. The recorded signals were amplified 20,000-fold for monkey T (8000-fold for monkey B) by a wideband preamplifier (MPA32I) and a programmable gain amplifier (PGA) (PGA 64, 1–5000 Hz; multi-channel systems GmbH, Reutlingen, Germany). The signals were referenced against the recording chamber, which was a titanium cylinder (25 mm inner diameter) in contact with the cranial bone and dura mater, and digitized using a 12-bit ADC at a sampling rate of 25 kHz. The ICM pulses were generated by an electric stimulus generator (STG 4008−1.6 mA, Multi-Channel Systems GmbH, Reutlingen, Germany). Each pulse was biphasic, consisting of 100 μs of cathodic current, followed by a 60 μs gap and 100 μs of anodic current. The strength of each pulse was ±25 μA for monkey B. For monkey T, a few initial recordings were performed with 15 μA pulses only, and 15 μA and 25 μA pulses were randomly applied for each trial for all subsequent recordings. A single pulse was provided at a random time within the second half of each MC (550−950 ms, Fig. 1C). For more details regarding the recording and stimulation setup and settings, see Drebitz et al.[42].

### Data analysis
Neuronal data were collected with the commercially available software MC Rack, v. 4.6.2 (release date: 2015-02-12; Multi-Channel Systems GmbH, Reutlingen, Germany). All data were analyzed with custom MATLAB scripts (version R2016b and R2020a, MathWorks, Natick, MA). Calculations and statistics with circular data were performed with the open-access MATLAB toolbox Circular Statistics Toolbox (v. 1.21.0.0, downloaded 06/14/2017)[64]. The stimulation artifacts in the recorded data were removed using a previously published method[42].

### Laminar electrode position
The stimulation electrode was lowered into the cortex before each recording session, and its location in the supragranular layers of V2 was verified based on the polarity of the average LFP onsets transient[65,66], which was evoked by a pattern reversal of a checkerboard stimulus.

**Table. 1 | The selection criteria for recording sessions and recording sites (several recording sites per session possible), are listed in the first column of the table and explained in the "Methods" section "Data selection"**

| Selection criteria | Applied in analysis | Samples passing criteria/total number of samples |
|---|---|---|
| Performance of at least 75% correctly performed trials | Applied to all analyses | Monkey B: 17 of 31 recording sessions monkey T: 40 of 48 recording sessions |
| (I) Number of misses is higher in ICM— than in non-ICM trials | Applied for all analyses except the comparison of miss-rate in ICM vs non-ICM trials (Fig. 2A) | Monkey B: 16 of 17 recording sessions, with 60 recording sites monkey T: 26 of 40 recording sessions, with 102 recording sites |
| (II) Average γ-band power in MC 2/3 70% above baseline power | Applied for all analyses except the comparison of miss-rate in ICM vs non-ICM trials (Fig. 2A) | Monkey B: 54 of 60 recording sites monkey T: 65 of 102 recording sites |
| (IIIa) location of the electrode tip in the granular layer | Applied for analyses of phase dependence of misses, RTs (Fig. 2 B/C), and the relation between spiking activity and RTs (Fig. 4) | Monkey B: 16 of 54 recording sites monkey T: 20 of 65 recording sites |
| (IIIb) Significant increase in spiking activity during stimulus presentation | Applied for analyses of the phase-dependent effect of ICM on ESA and LFP (Fig. 3) | Monkey B: 49 of 54 recording sites monkey T: 41 of 65 recording sites |

Each cell in this column represents a distinct criterion, and the corresponding entry in the second column provides the analyses to which the respective criterion was applied. In the third column, the first value indicates the number of sessions or sites that passed the respective criterion and the total number to which the criterion was applied, considering the consecutive order of application. Note that following criterion II, the analyses diverged into two branches: one with criterion IIIa applied, and another with criterion IIIb applied (with no requirement for samples fulfilling IIIa to fulfill IIIb, and vice versa. The Roman numbers are equal to those used in the main text to differentiate between criteria.

For acute V4 recordings, the laminar location of the electrode tip was assessed offline based on the polarity of the onset transients within 0–250 ms after visual stimulus onsets (Fig. 1C) and comparisons with laminar recordings (32 recording sites spaced by 100 μm, iridium-oxide coating, 0.25–0.35 MΩ Atlas Neuroengineering bvba, Leuven, Belgium). The V4 and V2 RFs were manually mapped in each session based on multi-unit and LFP responses to bar stimuli.

### Data selection

The initial analysis of the effect of ICM on the proportion of missed responses (independent of the γ-phase) is based on all recording sessions in which animals responded correctly in at least 75% of all trials (excluding trials with fixation failures, Table 1: row 1). Recording sessions and their recording sites used for all subsequent analyses fulfilled the following two additional conditions: (I) the proportion of misses in ICM trials had to be higher than in non-ICM trials (sessions fulfilling this criterion fall above the line of identity in Fig. 2A). (II) For each V4 recording site the average γ-band power (52–118 Hz, see Drebitz et al.[6]) during MCs 2 and 3 across successful trials had to be at least 70 % higher than that during the 800 ms interval beginning 70 ms after baseline period onset, Fig. 1C) in each of the two task conditions with a single, attended stimulus present within the V4 RF.

The data used for all subsequently listed behavioral and neurophysiological analyses came from successful trials (except when misses were required) of the two task conditions requiring selective attention for one of the two stimuli in the V4 RF. (IIIa) To examine the γ-phase dependence of the effect of ICM on the rate of misses and RTs and to analyze the relation between ICM-evoked spikes and RTs, we excluded recording sites outside the granular layer of area V4. This was necessary to prevent averaging across sites in different layers, as γ-oscillatory activities are laminar-specific and exhibit characteristic phase shifts across layers[55,66].

(IIIb) To analyze the effect of ICM on V4 neurons' spiking activity and the LFP depending on the phase of the local γ-oscillatory activity, we required a significant enhancement in the spiking activity in response to visual stimulation during MCs 2 and 3 compared to the spiking activity in the 800 ms interval beginning 70 ms after baseline period onset in both conditions with a single attended stimulus within the V4 RF (Wilcoxon rank sum test, $p \leq 0.05$).

Trials satisfying the above criteria provided up to four 650 ms segments of neuronal data from the V4 sites, which started 500 ms before each ICM pulse. To be included, each had to fulfill the following additional criteria: (1) more than 150 ms had to pass between the ICM pulse and the behavioral response. (2) To ensure a sufficiently large V4 γ-oscillation amplitude for reliable computation of the γ-phase 5 ms

after the ICM pulse in the stimulated trials (and 5 ms after the fictitious ICM pulse in trials without ICM used for generating surrogates; see "Statistical evaluation"), we required at that time a γ-power $P(t)$ of the LFP signal larger than the median of the 650 γ-power values along the segment. (3) To analyze the γ-phase-dependence of misses and RTs, data segments (and corresponding ICM pulses) had to come from the terminal MC of trials where animals either failed to respond or responded correctly. (4) To analyze the spiking activity, all (up to four) data segments around ICM pulses within a trial were used (provided that they fulfilled criteria (1) and (2)).

### Spiking activity

To analyze spiking activity in V4, we calculated the ESA[6,41,42]. After the removal of stimulation artifacts (39), the raw signal was high-pass filtered with a cutoff frequency at 400 Hz using a linear-phase FIR filter (stop band at 200 Hz, 70 dB suppression), applied bidirectionally, full-wave rectified, and low-pass filtered using a Gaussian kernel ($\sigma = 0.5$ ms, 2 ms window size, see Drebitz et al.[41] and Drebitz et al.[42] for details). For the illustration in Fig. 1D, a larger Gaussian kernel was used to calculate the neuronal responses at sites in V2 and V4 to each of the two stimuli presented individually within the V4 RF ($\sigma = 20$ ms, 60 ms window size). The resulting ESA of each recording site and condition was then z-scored by subtracting the mean ESA (over time and trials) during the baseline period of the respective trials (starting 100 ms after baseline onset and ending 100 ms before baseline end) and then dividing by the average of the standard deviations calculated for each time bin over trials within the baseline period. The time courses of the neuronal responses averaged across all recording sites for each of the two task conditions were normalized by dividing by the maximal ESA of both task conditions, separately for each animal.

### LFPs and phase estimation

To estimate the phases of the V4 γ-oscillations, the raw signals (1–5000 Hz, 25 kHz sampling rate, electrical artifacts removed[42]) were band-pass filtered bidirectionally in a broad γ-frequency range (35–125 Hz) using an equiripple finite impulse response (FIR) filter (stop frequencies at 20 Hz and 150 Hz, stopband suppression at 60 dB and 80 dB) and downsampled to 1000 Hz. Subsequently, the γ-phases $\Phi_{LFP}(t)$ and amplitudes $A_{LFP}(t)$ were estimated using a Hilbert transform[67,68]. Due to differential conduction delays, the impedance of the recording electrode, and the differences in the tip orientations of the electrodes within the recorded population, the LFP's γ-phase is shifted differently for different recording sites compared with the phase of the γ-oscillatory excitability cycle of the recorded population[69,70]. To correct this and express the phase values in terms of

the excitability cycle, we shifted the γ-phases derived from the LFP such that within the average γ-cycle, the maximum ESA occurred at 0° (see also Lisitsyn et al.[52]). To this end, we used the ESA-signal and LFP γ-phase values during MCs 2 and 3 of all trials with and without ICM, except for the periods between 20 ms before and 50 ms after an ICM pulse to compute the means of the ESA values sorted into 15° bins according to the γ-phase of the LFP for each V4 recording site and fitted the resulting histogram with the function $y(\varphi) = a*\cos(\varphi + b) + c$ (with $a \geq 0$, $-\pi \leq b \leq \pi$, and $c \geq 0$). Then, the phase shift $b$ was added to the recording site's phase values of the LFP to obtain $\Phi_{ExC}(t)$, the γ-phase of the excitability cycle at times $t$.

To analyze the phase dependence of the behavioral and physiological consequences of ICM pulses, we used the excitability cycle's γ-phase $\Phi_{ExC}(t)$ with $t$ set to 5 ms after the ICM pulse to obtain reliable phase values unperturbed by ICM-evoked activity in V4 or potential remnants of the electrical artifact. The γ-phases at later time points of interest (10 ms and 9.2 ms) were calculated based on $\Phi_{ExC}(t)$ at 5 ms, the time difference between 5 ms and 9.2 ms or 10 ms, and the animals' average γ-peak-frequency (Supplementary Fig. 2 and Supplementary Notes 1). The latter was calculated based on the mean phase progression of $\Phi_{ExC}(t)$ between 5 ms before and 5 ms after the ICM pulse across all selected data segments for each animal.

To test whether the phase estimate at $t = 5$ ms might be substantially influenced by ICM-evoked responses starting approximately 5 ms later (Fig. 3A, B), we simulated the effect of the observed responses on γ-phase estimates. For this, we used the same raw data segments (25 kHz sampling rate, electrical artifacts removed) already used to analyze the γ-phase dependent effects on ESA and LFP, but during periods preceding the actual time of phase estimation by 200 ms, which precedes any possible ICM-evoked effect on γ-phase. Each segment was required to fulfill the same criterion for γ-power at the time of γ-phase estimation (here 195 ms before ICM) as used in the actual analysis (see "Methods" section: "Data selection", criterion 2). The selected data was then processed in two parallel streams.

(1) In the first stream, each segment was filtered bidirectionally using the broadband γ-filter described above, and the γ-phase at $t = -195$ ms was estimated via Hilbert transformation, following the original procedure for phase estimation. (2) In the second stream, the same raw data segments were used to simulate the effect of ICM-evoked LFP responses on γ-phase estimation. For this, we used as a template the largest average ICM-evoked response observed in the LFP (Fig. 3B, top panel, blue curve), defined as the negative deflection occurring between 9.72 ms and 22.12 ms after ICM onset. This response template was added to each data segment at the same relative time with respect to the γ-phase estimation time (i.e., from −190.28 ms to −177.88 ms) as at the actual estimation time, 5 ms after the ICM-pulse, thereby preserving the original temporal relationship between response onset and phase estimation. The γ-phase at $t = -195$ ms was then estimated using the same filtering and Hilbert-based method as applied to the original data. To test the influence of stronger responses, the same procedure was repeated using response templates scaled by factors of 5 and 10. The γ-phase values obtained from both processing streams were compared to quantify the potential influence of ICM-evoked LFP responses on γ-phase estimation (Supplementary Fig. 9).

### Phase dependence of the missed responses, RTs, LFP, and ESA

First, we investigated whether the occurrence of missed responses depended on the γ-phase associated with the ICM pulse during the MC requiring the behavioral response. For each selected recording site in both animals, we constructed a histogram providing the number of misses across γ-phases at 9.2 ms after the ICM pulse with bins of 10° width. Each bin's count was normalized by dividing by the number of

trials with γ-phase values in the bin's phase range, resulting in correct responses or misses. The results were smoothed with a Gaussian kernel ($\mu = 0$, $\sigma = 30°$, steps of 5°) and subsequently averaged by calculating the binwise median across recording sites, finally obtaining the average missed response rate as a function of the γ-phase $\overline{mrr}(\varphi)$.

To determine the γ-phase dependence of the RTs, we first subtracted the median RT for each session across all successful trials without ICM from the RTs of successful trials with ICM, which provides the excess response times (ERTs). Next, for each recording site, we computed $ert_s(\varphi)$, the excess response time as a function of the γ-phase $\varphi$ in 5° steps using Gaussian weighted averaging:

$$ert_s(\varphi) = \frac{\sum_{i=1}^{n}\{ERT_i * g(\varphi - \Phi_{ExC}(9.2))\}}{\sum_{i=1}^{n}\{g(\varphi - \Phi_{ExC}(9.2))\}} \qquad (1)$$

where $n$ is the number of ERTs and the corresponding data segments fulfilling the selection criteria (see above), $\Phi_{ExC}(9.2)$ is the γ-phase 9.2 ms after the ICM pulse, and $g(\varphi)$ is a Gaussian function with $\mu = 0$ and $\sigma = 30°$. For each animal, we computed $ert_a(\varphi)$ for each $\varphi$ as the median across all $ert_s(\varphi)$.

The ranges around ±45° of the maximum and minimum of $ert_a(\varphi)$ or $\overline{mrr}(\varphi)$ define the effective (EP) and ineffective phase range (IP), respectively, for statistical analyses.

A similar approach was used to determine the γ-phase dependence of the spiking activity. The function $esa(t, \varphi)$ provided the average ESA across the selected recording sites of both animals, depending on the time $t$ passed after an ICM pulse and the γ-phase $\varphi$ at that time:

$$esa(t, \varphi) = \frac{\sum_{i=1}^{n}\{ESA_{i,t} * g(\varphi - \Phi_{ExC}(t))\}}{\sum_{i=1}^{n}\{g(\varphi - \Phi_{ExC}(t))\}} \qquad (2)$$

where $n$ is the number of all selected data segments, $ESA_{i,t}$ is the ESA value at time $t$ of data segment $i$, $\Phi_{ExC}(t)$ is the γ-phase of the excitability cycle at time $t$ in data segment $i$, and $g(\varphi)$ is the Gaussian weighting function with $\mu = 0°$ and $\sigma = 30°$. This value is independent of the animals' different γ-band frequencies and, therefore, allows the pooling of ESA data across animals (for illustration, see Supplementary Fig. 7). Using this function, we calculated $esa_{tc}(t, \varphi_{t_{ref}}, t_{ref})$, which is the average time course of the ESA following ICM pulses with the same γ-phase $\varphi_{t_{ref}}$ at time $t_{ref} = 0.01$, i.e., at 10 ms after the ICM pulse around the response onset:

$$esa_{tc}(t, \varphi_{t_{ref}}) = esa(t, \varphi_{t_{ref}} + 2\pi \times f_{mean} \times (t - t_{ref})) \qquad (3)$$

We then convolved this value with a 2D Gaussian kernel for smoothing (temporal domain: $\sigma = 0.12$ ms, window size: 0.52 ms; phase domain: $\sigma = 15°$, window size: 65°), where $f_{mean}$ is the mean γ-band frequency of the animals (73.5 Hz).

We used multiple ($n = 15{,}000$) surrogate data sets composed of trials without ICM and statistical and temporal parameters matching the data set with ICM (for details, see "Statistical evaluation", Supplementary Fig. 7) to calculate the mean ESA values

$$\overline{esa}_{tc,noICM}(t, \varphi_{t_{ref}}) = \frac{1}{n}\sum_{i=1}^{n} esa_{tc,noICM,i}(t, \varphi_{t_{ref}}) \qquad (4)$$

and the standard deviations

$$SDesa_{tc,noICM}(t, \varphi_{t_{ref}})$$
$$= \sqrt{\frac{1}{n-1}\sum_{i=1}^{n}\left(esa_{tc,noICM,i}(t, \varphi_{t_{ref}}) - \overline{esa}_{tc,noICM}(t, \varphi_{t_{ref}})\right)^2} \qquad (5)$$

across the $n$ surrogate data sets. The $z$-scored ESA response (in Fig. 3A) was obtained as follows:

$$Resa_{tc}\left(t,\varphi_{t_{ref}}\right) = \frac{esa_{tc,ICM}\left(t,\varphi_{t_{ref}}\right) - \overline{esa}_{tc,noICM}\left(t,\varphi_{t_{ref}}\right)}{SDesa_{tc,noICM}\left(t,\varphi_{t_{ref}}\right)} \quad (6)$$

The procedure to determine the γ-phase-dependent effects of ICM pulses on the LFP was identical to the procedure outlined for the ESA, including the statistical analysis, except that the surrogate data set contained 10,000 samples. The only distinction lies in the data used: for the LFP, we employed the artifact-removed raw signal without the additional processing steps required to calculate the ESA.

### Relationship between ICM-evoked spiking activity and RTs

To investigate whether there might be a direct link between the additional V4 spikes caused by ICM and the RT delays, we examined how RTs varied depending on the success or failure of ICM pulses in causing additional spikes in V4. To (1) define thresholds for spike detection unperturbed by ICM and (2) to determine the number of spikes expected without ICM pulses, we first configured a surrogate dataset based on the high-passed raw signal (same filter settings as in "Methods" section: "Spiking activity") of trials without ICM and the time points of actually occurring ICM pulses within trials with ICM.

For compiling this surrogate dataset, we used only those ICM pulse times that served to estimate spiking activity after an ICM pulse (Table 1 IIIa). Of these pulses, only the last before the behavioral response and associated with the effective phase range (−15° ± 45°) at 9.2 ms after the ICM were taken into account. Each of these pulse times (template) was then randomly assigned to one of the trials without ICM of the same recording site to position a fictitious ICM pulse. This assignment was constrained to trials without ICM that were sufficiently long to accommodate the respective fictitious ICM pulse and a subsequent period of at least 150 ms before the behavioral response, consistent with the constraints applied to analyses of ICM trials. Within a 10 ms window (centered at 5 ms after the fictitious ICM pulse), we identified the time point with the smallest difference between its γ-phase and the γ-phase at 5 ms after the actual pulse in the template trial. The fictitious ICM pulse was then repositioned to 5 ms before this time point, and the data within the subsequent 6 ms window, starting 9.2 ms after the fictitious pulse, was used later on to estimate the threshold for spike detection and the expected number of spikes within this period. This procedure was repeated 1000 times for each recording site, resulting in the final surrogate dataset.

(1) To define an unperturbed spike detection threshold for each recording site based on the statistical characteristics comparable to those of the signals following the ICM pulses, we calculated the standard deviation of the signal values within each 6 ms surrogate data segment of a given site and averaged them. The recording site's threshold was then determined based on spike polarity. If the average positive signal components exceeded the absolute average of the negative components, the threshold was set to $+(2 \times \overline{STD})$ and otherwise to $-(2 \times \overline{STD})$.

(2) To determine the number of spikes that are expected to occur in the 6 ms windows from 9.2 ms to 15.2 ms after the ICM pulses, if there were no ICM pulses, we again used the surrogate dataset based on trials without ICM. We counted how often the signal crossed the recording site-specific threshold within each of the 6 ms windows following the fictitious pulses for each recording site. This provided a distribution of spike counts for each site, which was used to calculate the expected number of spikes by taking the mean across all spike counts for a given site.

To determine whether, in trials with ICM, an ICM pulse associated with the effective γ-phase range (−15° ± 45° at 9.2 ms) evoked

additional spikes in V4, we first counted the spikes that crossed the recording site-specific threshold within the 6 ms time window between 9.2 ms to 15.2 ms after the pulse and then subtracted the recording site's expected number of spikes for a 6 ms window. The spike counts resulting in differences within the range of [−1 to 1] were classified as average, while those resulting in differences > 1 were classified as above average. The same procedure was performed to classify the spike counts within corresponding 6 ms time windows that follow the fictitious ICM pulses in the surrogate dataset into an above-average and an average group.

To assess changes in RT due to additional V4 spikes that were caused by ICM pulses associated with the effective γ-phase range, we subtracted from the group of RTs following ICM pulses that resulted in above-average spike counts the median of the RTs following fictitious ICM pulses in the recording site's surrogate dataset that resulted in above-average spike counts and compared them with the group of RTs following ICM pulses resulting in average spike counts after subtracting the median of the RTs following fictitious ICM pulses in the recording site's surrogate dataset that resulted in average spike counts.

### Comparison of the γ-phase dependent effect of ICM for different frequency ranges

To investigate whether the bandwidth of the broadband γ-filter used to obtain the broadband γ-LFP covered the relevant frequency ranges and did not bias the results towards a specific γ-frequency band, we compared the effect of ICM on RTs for different frequency bands, covering also the β- and high γ-frequency bands (Supplementary Fig. 8). The raw data (25 kHz sampling frequency, artifacts removed) were band-pass filtered between 17 Hz and 110 Hz bidirectionally for this analysis. The FIR filter had a lower cutoff frequency at 12 Hz with an attenuation of −20 dB and an upper cutoff frequency at 115 Hz with an attenuation of −15 dB. The passband ripple was constrained to 0.5 dB.

The subsequent procedures were analogous to those used with the broadband γ-filter, as detailed in the "Methods" section: "LFPs and phase estimation". Briefly, the Hilbert transform of the band-pass-filtered data provided the instantaneous phases, which we also used to calculate the instantaneous frequency around the ICM pulses. Specifically, the average phase progression in a 20 ms window centered around the time of phase estimation (−5 ms to 15 ms relative to the ICM pulse) was used to determine the instantaneous frequency during that period.

To compare the phase-dependent effects of ICM on RTs across different frequencies, RTs were grouped according to their associated instantaneous frequencies in 20 Hz ranges, starting with 20 Hz to 40 Hz and increasing in steps of 5 Hz, up to the final range of 90–110 Hz. For each frequency range, RT-modulation curves were calculated for each recording site, and median RT-modulation curves were used to identify the animal-specific effective and ineffective phase ranges (±45° around the maximum and minimum, as described in detail in "Phase dependence of the missed responses, RTs, LFP, and ESA"). The RTs associated with the animal-specific effective and ineffective phase groups were both pooled across sites and animals for each frequency range separately. The difference between their median RT associated with the effective phase range and the median RT associated with the ineffective phase range provided the spread for a given 20 Hz frequency range (see Supplementary Fig. 8). Note that a minimum of five phase values within a given frequency range was required to calculate a session's RT spread for this frequency range. Due to the low number of cases with frequencies below 35 Hz and above 100 Hz, these frequency ranges could not be analyzed.

The statistical evaluation was based on shuffle controls for each frequency range. RTs were randomly reassigned to the observed phases within a given recording site and frequency range. The subsequent procedures were identical to those described above, yielding median RT-modulation curves for each animal based on randomized

RT-to-phase relations. The RTs falling within the effective and ineffective phase ranges obtained from these curves were then pooled across recording sites and animals for each 20 Hz frequency range, and the spread between the median RT of the effective and the median RT of the ineffective phase ranges was calculated. This procedure was repeated 5000 times for each 20 Hz frequency range.

### ICM-induced changes in gaze direction

To assess whether the single ICM pulses induced changes in the direction of gaze, including microsaccades or drifts (Supplementary Fig. 1), which may explain the behavioral effects associated with ICM pulses, we first filtered the recorded analog eye position signals. We applied a low-pass FIR filter bidirectionally (250 Hz pass, 300 Hz stop at 25 dB suppression, passband ripples were restricted to ±0.025 dB), and subsequently downsampled the signals to 1000 Hz.

For each trial, we analyzed the time course of the direction of gaze between 100 ms before and 150 ms after ICM pulses. Since eye movements following ICM in V1 have a latency of at least 50 ms applied in the upper cortical layers[71], we used the mean gaze position in the first 50 ms after the ICM pulse as a reference position. To obtain the time course of the deviation of the gaze direction from this reference position, we computed the Euclidean distance between the reference position and the current position of gaze for each time bin following the reference period (50–150 ms post-ICM pulse). This calculation was performed for the 100 ms preceding the reference period, which allowed us to assess the expected variability of comparable data in periods not influenced by ICM.

To investigate whether ICM pulses resulted in potential γ-phase dependent changes in the direction of gaze, we averaged the time courses of gaze deviation following the methodology outlined in the "Methods" section: "Phase dependence of missed responses, RTs, LFP, and ESA". This procedure resulted in a matrix containing the median deviation of the gaze direction from the reference direction as a function of time around an ICM pulse and γ-phase for each recording site. These matrices were averaged across recording sites for each individual animal.

Because of the bias that the average deviation of gaze direction from an initial reference direction becomes progressively larger, we generated a surrogate dataset to subtract this bias. To create the surrogate dataset for a recording site, a random time point between 500 ms and 400 ms before a randomly chosen actual ICM event was selected. The Euclidean distance between the average direction of gaze in the 50 ms reference period preceding this random time point and the gaze direction values in the subsequent 100 ms was calculated. This procedure was repeated as many times as the number of actual ICM pulses from each recording site. The resulting 100 ms periods of gaze direction deviation values were then randomly assigned to an actual γ-phase value for that recording site, and the γ-phase dependence matrix was calculated as described above. For each animal, the resulting matrices were averaged across recording sites. We replicated this procedure 1000 times, and the mean across these iterations was used to subtract the inherent bias resulting from increasingly larger deviations in gaze direction over time. Additionally, the surrogate dataset was used to assess potential significant changes in eye position induced by the ICMs (see Supplementary Fig. 1).

### Statistical evaluation

The significance of the difference between the proportion of missed responses in all trials (excluding fixation errors) with and without ICM was examined with Wilcoxon signed-rank tests.

To test whether the observed number of missed responses associated with a V4 γ-phase (at 9.2 ms after the ICM pulse) within the EP (or the IP) deviates from the null hypothesis, we calculated the expected number of missed responses $n_{expmiss}$ within these two phase ranges for each session. Under the null hypothesis of no relationship between the

γ-phase and the effectiveness of ICM pulses in eliciting response failures, the phase values associated with missed responses are distributed uniformly across the γ-phases. Using $\widetilde{Q}_{miss,noICM}$, the median of the selected recording sites' proportions of missed responses among trials without ICM that end with a correct or missed response (12.5%), the expected number of missed responses among trials with ICM is given by:

$$n_{exp\,miss} = (n_{miss} + n_{corr}) * \widetilde{Q}_{miss,noICM} * 90/360 \tag{7}$$

The significance of the difference in the number of missed responses observed for the EP (±45° around the maximum) and IP (±45° around the minimum, therefore 90°/360 ° in the equation) was tested against the expected numbers within 90° phase ranges using Wilcoxon signed-rank tests.

To test whether the modulation depth of RTs and misses as a function of the γ-phase was statistically significant (Fig. 2B, C), we performed shuffle controls. This procedure randomly reassigned the RTs and the correct responses and misses to the associated γ-phases for each recording site. Then, the modulation curves were obtained as described in the "Methods" section: "Phase dependence of the missed responses, RTs, LFP, and ESA", and the curves' peak-to-peak amplitudes (difference between the minimum and maximum) were calculated. This procedure was repeated $n_{shuf} = 10,000$ times to generate a distribution of values representing the chance level. This distribution was used to estimate the probability that the observed peak-to-peak amplitudes of RTs and missed responses following ICMs were significantly different from chance.

The statistical significance of ICM-evoked modulations of spiking activity and the LFP was assessed by comparing them with multiple sets of surrogate data constructed from trials without ICM (but with fictitious pulse times). The selection criteria for trials and their 650 ms data segments (around fictitious ICM pulses) and all processing steps were the same as those for the trials and data segments with ICM.

A single surrogate data set was created using all ICM pulses and their surrounding data segments (which were used to calculate the ESA and LFP response) as templates. For each template, we randomly selected from the same recording site a trial without ICM that was sufficiently long to contain a fictitious pulse with the surrounding data segment at the time the template's ICM pulse occurred in the trial. Then, we compared the surrogate segment's γ-phases between −5 ms and +15 ms around that time with the template's γ-phase 5 ms after the ICM pulse and shifted the fictitious pulse to the time 5 ms before the time at which the difference from the template's γ-phase was minimal. If this surrogate data segment did not meet the selection criteria (see "Data selection"), the procedure was repeated, starting with the random selection of another trial without ICM from the same recording site. The resulting pool of surrogate data segments contained the same number of data segments at essentially equal times within trials and with equivalent γ-power and γ-phase characteristics as the pool of data segments with actual ICM. This compilation of surrogate data sets was performed 15,000 times for ESA and 10,000 times for the LFP analysis (see "Methods" section: "Phase dependence of the missed responses, RTs, LFP, and ESA"; and Supplementary Fig. 7).

We tested the statistical significance of the γ-phase-dependent ESA and LFP responses to ICM application based on the probability of finding similar or larger responses in the surrogate data sets. We first averaged the ESA and LFP values in a 2 ms window around the identified peak in the ICM data for each of the 15,000 (10,000) surrogate data sets and identified the maximum (minimum for LFP) across phases for each set. These maxima of all surrogates were used to construct the probability distribution. We then derived the likelihood $p$ of an event with at least the maximum ESA and LFP response observed after ICM application based on these distributions (bootstrap hypothesis testing[72]).

To quantify whether the ICM-evoked ESA response depends on the $\gamma$-phase, we calculated the nSVs (8) $\text{nSV} = \sum_{i=1}^{n} \vec{A}_i / \sum_{i=1}^{n} |\vec{A}_i|$ across vectors $\vec{A}_i$ with directions $\varphi_i$ and magnitudes equal to the mean (of the 2 ms window) $z$-scored ESA values in phase $\varphi_i$ for each of the $n$ phase bins. The nSV is in the range of $0 \leq \text{nSV} \leq 1$. An nSV value close to 1 indicates that the ESA has a strong gamma phase dependence, while an nSV value close to 0 indicates no phase dependence. The nSV was calculated for each surrogate data set to construct a probability distribution of nSV values. This distribution was used to assess the likelihood $p$ of observing an nSV value equal to or larger than the nSV value for the ICM-evoked ESA response (bootstrap hypothesis testing[72]).

### Reporting summary
Further information on research design is available in the Nature Portfolio Reporting Summary linked to this article.

### Data availability
The data supporting this study are available from the corresponding author (drebitz@brain.uni-bremen.de) upon request, as they are large, complex, and context-dependent. Source data are provided with this paper.

### Code availability
The custom-made MATLAB scripts are available from the corresponding author upon request.

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

## Acknowledgements

This work was funded by the Deutsche Forschungsgemeinschaft (DFG, German Research Foundation)—331514942 and 238990875 to A.K.K. In addition, we thank Aleksandra Nadolski, Peter Bujotzek, and Katja Taylor for training and valuable technical assistance. We would also like to acknowledge Katrin Thoß and Ramazani Hakizimana for their diligent animal care. Furthermore, we express our appreciation for the insightful discussions with Rasmus Roese, Bastian Schledde, and Detlef Wegener on experimental procedures and equipment.

## Author contributions

E.D. made contributions to data acquisition, methodology, performed data analysis, and played a substantial role in data interpretation and visualization, as well as drafting the article. L.-P.R. contributed to data acquisition and initial data analysis. A.K.K. provided the conceptual framework and theoretical background for this work, contributed to methodology, data interpretation, and participated in manuscript drafting substantially.

## Funding

## Competing interests

The authors declare that they have no competing interests.
