## [Transparent Peer Review file · Nature Communications]

Gamma-Band Synchronization between Neurons in the visual cortex is causal for effective Information processing and Behavior

Corresponding Author: Dr Eric Drebitz

Version 0:

Reviewer comments:

Reviewer #1

(Remarks to the Author)

The elegant manuscript "Gamma-band synchronisation between neurons is causal for effective information processing and behaviour" by Drebitz, Rausch and Kreiter carefully investigates the impact of gamma-phase on the efficacy of an artificial signal to impair a visual perceptual task and evoke spikes in a downstream area. In this study, the authors inserted single pulses of electrical stimulation in extrastriate visual area V2 during a shape tracking task. They observed missed responses and prolonged reaction times dependent on the gamma-phase in V4 representations with overlapping receptive fields. Also, the ability of the artificial stimulation to evoke spikes in V4 was dependent on the V4 phase at the time of the supposed incoming signal.

This study provides important causal evidence for gamma-oscillations as neural mechanism for the effective information processing between visual areas. Data and methods are convincing and well described. There are a couple of areas where some clarifications or additional analyses would be helpful:

1) Strictly speaking the authors do not show the direct link between the additional spikes generated in V4 and changes in perceptual performance. However, I agree that this is a likely conclusion from the study. I appreciate that the data set and effect might be too small for this direct link, but could the authors report the misses and RT for the trials, for which they could record microstimulation-evoked spikes in V4?

2) Also, the authors assert in the first part of the paper that the gamma-phase in V4 matters for the behavioural effect but not the gamma-phase at the stimulation location in V2. Since the authors record also in V2, could they just show the analysis for V2 at the time just prior to stimulation?

3) The main behavioural effect is a disturbance of task performance in the form of misses. Therefore, I presume this effect would be less specific with regards to the visual representation than if there were an increase of specific false positives. Apart from the receptive field location, do the authors have any further data on the shape selectivity or the level of stimulus-evoked response of the recorded/stimulated V2 and V4 neurons and how these properties might be related to the probability of a microstimulation effect on task performance?

Such data could provide further insights into the neuronal mechanisms that underlie the perceptual coding in these tasks. The authors speak of the possibilities of evoking phosphenes, which of course would interfere with perception in this task, but could be pretty non-specific to the shape processing. Perhaps the authors would like to speculate more in the discussion about how they envisage the artificial micro-stimulation signal interacts with the shape representations that form the perceptual basis for this task. And perhaps furthermore, what it would take to alter the percept in a meaningful way, now they have an effective mechanism to insert spikes?

4) In the methods under data selection, the authors lay out a number of selection criteria for the different analyses they have

conducted. At the moment, the statements are somewhat ambiguous as to whether and at what stage there was some active pre-selection of sessions based on certain expected effects. For instance at the top of page 18, (I) states that for the data shown in Figure 2, only sessions with more misses on ICM than non-ICM trials were included. Then Figure 2A is referenced, which of course includes other sessions, too. Also, it was not entirely clear to me from the results text alone that some of the analyses have been done on subsets of sessions/trials. This is not a problem per se, but should be explicitly stated in the results where applicable.

It would also be helpful to the reader to give an indication of the number of excluded sessions at each stage of the analysis, starting from the initially recorded sessions. This could be done in the results in the text or perhaps as a table in the methods, referencing each Figure/Analysis.

Overall, this is an exciting, carefully conducted study that sheds light on a potentially important neuronal mechanism for information processing in the primate brain.

Reviewer #2

(Remarks to the Author)

This is potentially an outstanding study. The topic is of great importance, the experiments are very advanced and impressively executed, and the analysis is likely sound, even though it will be crucial to address some concerns listed below.

The authors recorded in V4 of awake macaques performing a very demanding selective visual attention task, while simultaneously applying intracortical microstimulation (ICM) to V2 sites. The V2 site had an RF that partly overlapped with the RF of the V4 site. Stimuli were placed such that two stimuli fell into the V4 RF and one into the V2 RF. Accomplishing these experiments is a veritable feat and the authors deserve the greatest respect! The experiments allow them to address a crucial question, namely whether ICM evoked inputs to V4 have a gain that depends on the V4 gamma phase. The results suggest that this is the case, and the ICM-evoked input affects V4 spiking and interferes with behavior in a strongly V4-gamma-phase dependent manner.

The results look very convincing. However, a few aspects of the analyses require clarification. If these concerns can be addressed convincingly, I can strongly recommend the manuscript for publication.

Major points:

Fig. 2B, C: I assume that the x-axes show the V4 LFP gamma phase after shifting it according to the V4 ESA, such that phase zero has maximal ESA. If so, the fact that the behavioral data, i.e. the y-axis values, peak at zero is a finding. Alternatively, the y-axis values have been realigned such that their peak is at zero. The legend states "The curve's maximum is centered at 0°", and it is not clear to this reviewer, whether this describes a finding or an operation performed by the authors during data analysis. If the authors centered the curve's maxima at zero, it would be trivial that the values decline for other x-axis values (this would be particularly concerning, if the centering would have been done per session, before averaging over sessions). Please clarify.

Line 360: "The regions $\pm 45^\circ$ around the maximum and minimum of $eeeeea(\varphi\varphi)$ or $mmmmmm(\varphi\varphi)$ define the effective (EPR) and ineffective phase range (IPR), respectively, for statistical analyses". By doing it like this, the authors essentially show merely that the dependent variable shows some dependence on the independent variable. In principle, for their statistics to become significant, the curves shown e.g. in Fig. 2B,C could take any shape, as long as the maximum and minimum have sufficiently different y-values; the maximum and minimum could e.g. occur at very nearby x-values, and the curve could be very different from a cosine. By contrast, the curves shown e.g. in Fig. 2B,C could probably be fitted well by a cosine, or a von-Mises function. I suggest that the authors consider using this observation, because it makes for a much stronger case. Also, some readers might consider it problematic if the authors first select maxima and minima, and then show that they are different. I think that the observed differences are not trivial, but I also think that the results support stronger inferences. At the least, the authors could simply fix the EPR to be centered at zero, and the IPR to be centered at 180 degrees, i.e. to be defined by the ESA. This would avoid concerns about circularity (assuming that zero phase is defined solely on the basis of the ESA – see my previous comment).

Similar concerns apply to the analysis that first selects the maximum ESA response and then compares it to a surrogate distribution (line 150). Some readers might perceive this as circular. I think it is not circular, it is just an analysis that is less strong than it could probably be, given the presented results. I find more elegant the analysis presented in lines 156-159, which avoids the selection of the maximum.

The authors use three different delays after the ICM pulse for their analyses:

Line 144: "These gamma phase were estimated at 10 ms after the ICM pulse"

Line 336: "To analyze the phase dependence, we used the γ -phase $\Phi LLLLLL \varphi\varphi\varphi$ with $\varphi\varphi\varphi$ set to 5 ms after the ICM pulse."

Line 346: "shared γ -phase at 9.2 ms after the ICM pulse that fell into the same 10° phase bin". How were the values of 5 ms,

9.2 ms, and 10 ms chosen? Why were there three different values? Also, I do not understand what is meant by "that fell into the same 10° phase bin"; please clarify. These questions are partially answered around line 170, but the authors should explain this earlier and in more detail.

Further points:

Line 35: „Essentially, neurons respond as if only the attended stimulus is present“. I suggest to specify that this holds for neurons in higher areas, with two competing stimuli in their RF.

In Fig. 1B, the blue RF and labeling seems to have a low contrast relative to the grey background.

In the legend for Fig. 2B, I found this description confusing: "The graph shows the median proportion of MRs between correct and missed responses". The methods states: "percentage of trials resulting in missed responses among all trials that resulted in correct or missed responses", which is much clearer.

Line 228: I presume that the cue, i.e. the annulus, disappeared before the shapes appeared. Please clarify.

Line 330 refers to "Lisitsyn et al., 2020", whereas the referencing style and the bibliography are numbered.

The authors shift the LFP phase estimates to align maximal ESA to phase zero. To clearly label this shifted LFP phase as such, they might consider to refer to it throughout as "corrected", "shifted", or "aligned" phase, or similar, and also use a respective abbreviation for it.

Fig. 2B,C: I guess that B shows the pooled data of both monkeys, and C shows the individual data per monkey. Please clarify in the figure legend or by respective labeling of the figure panels.

Line 88 and Stable 1: It seems that ICM reduced the false alarm rate in monkey T. This should be mentioned and discussed.

Line 124: This control is extremely elegant and convincing. Can the authors show for the present data the effect of attention on V2-V4 coherence, which is described in lines 120-123?

Reviewer #3

(Remarks to the Author)

In the manuscript "Gamma-Band Synchronization between Neurons is causal for effective Information processing and Behavior" Drebitz et al. stimulated in rhesus monkey area V2 with single low amplitude single biphasic pulses while animals performed a demanding selective attention task, and recorded spiking and field potential activity in area V4. The V4 site had receptive fields that included two stimuli, while the V2 site contained either the attended or unattended of those stimuli.

The study reports two major novel key findings. The stimulation pulse induces an apparent strong behavioral effects, causing the two animals to miss the target shape reoccurrence and slowing reaction times (Fig. 2). The behavioral effect was evident selectively at a narrow gamma band phase of the receiving V4 site. Secondly, the authors demonstrate that the V2 stimulation had effects on the spiking response in area V4 at a narrow gamma phase range (gamma measured in V4) and temporal lag relative to the V2 stimulation (Fig. 3).

The authors performed critical control analyses by assigning 'sham' stimulation pulses and analyzing their effects as a function of the gamma phase, observing no phase dependent changes in behavior.

The paper reports an outstanding novel finding of broad significance, is very well written, the statistical tests appropriate and the reporting consistently shows effects in both animals. The authors provide extensive supplementary information that illustrate the careful selection of recording sites and that the gamma phase can be meaningfully be estimated, amongst others.

Overall, this is an exceptional manuscript with a concise and highly innovative insight about the causal role of the postsynaptic, recipients gamma phase on actual spiking responses and behavioral output using highly subtle, low amplitude perturbations. This result likely is a direct consequence of a careful and sophisticated experimental set up and careful temporal analysis of the temporal lagged phase dependent response.

There are aspects of the manuscript deserving considerations to strengthen the possible inferences from the results and the comprehensiveness of the results, as suggested below.

- The interpretation of the authors point to the gamma phase as the key determinant of the evoked spiking and behavioral effect. It would clarify the frequency specificity if the authors could identify a frequency range (towards low beta or higher gamma) at which the phase dependency of the effect disappears. This could also help future studies when identifying the frequency at which phase estimates can meaningfully predict stimulation effects.

- The microsimulation was delivered during the 1st, 2nd, 3rd, or 4th morphing cycle. Can the authors add information clarifying that the relative time of the stimulation pulse does not

change the main effect,, i.e. that the phase-dependent evoked field amplitude (or the behavioral misses) is qualitatively similar for early and later stimulation pulses. A difference of the evoked field would not question the papers key results, but it could help interpreting the effect as being linked or not linked to attentional expectancy.

- It is difficult to discern for a reader whether the stimulation effect on V4 responses varied between electrode contacts from different layers of the V4 site. Is it possible to distinguish layers, or upper versus lower recording sites, to constrain the interpretation of the findings to be potentially stronger for the upper V4 layers ?

- It is explicitly acknowledged that the authors are focusing on the (entire average) spiking responses as their metric evaluating stimulation effects, which is an exceptional success. I can imagine that the readership, however, will include human noninvasive EEG researchers, who will wonder whether the evoked stimulation effects are not also, or even stronger and longer, showing up in the local field potential measure. Describing whether there was any hint of an evoked field effect could enhance the impact of this paper to a wider audience. It would be particularly interesting if there would not be any stimulation triggered LFP effect and only the transient spiking effect, because it will reveal that the authors found the right modality to reveal causality of phase dependent spiking responses. The results is ideally mentioned explicitly in the manuscript.

- The success of the manuscript seem to rely on the high sophistication to place the stimuli so that the V2 and V4 sites are jointly stimulated and an additional stimulus is present in the V4 receptive field. But results and illustrations of this important aspect are not included in this manuscript. The reader will benefit to see more explicitly that the V2 sites and V4 sites do show onset responses to the stimulus. Alternatively, the author may want to show cases where phase dependent stimulation effect does not occur because of suboptimal placement of stimuli. Adding more information on this would make the particular strength of the papers approach more apparent to readers.

- Please also mention explicitly that the microstimulation pulse did not (or did) have effects on the gaze position. One possible scenario to account for the behavioral effect could include a stimulation effect to offset the foveation point (and/or cause a phase reset), disrupting upstream areas. This is ideally mentioned/discussed explicitly in the text, given evidence in the literature that phases or power of visual gamma activity maybe linked to microsaccadic movements.

- The title of the manuscript would benefit from explicitly mentioning that the results are from visual cortex. There are still open questions on how important gamma phases are in other brain systems (with less focal or dense connectivity) and the paper may not want to insinuate the same effects are independent of the brain system.

Version 1:

Reviewer comments:

Reviewer #1

(Remarks to the Author)

I am content with the clear answers and the additional analyses the authors have provided and have no further questions.

Reviewer #2

(Remarks to the Author)

The authors have addressed my previous comments by their revisions and responses. However, their responses raised one new concern that should be addressed. In response to my main point #4, the authors wrote that phase values around 9-10 ms could not be taken directly from the signal, because it might already be influenced by an evoked response. I agree with the authors that the phase estimation should exclude parts of the signal that might already be influenced by an evoked response. However, is this accomplished by their current phase estimation procedure? This procedure first removes the electrical artifacts, then applies a broad band-pass filter in the gamma range bidirectionally and finally uses the Hilbert transform to obtain the gamma phases and amplitudes. The filter is a FIR filter, which is by construction acausal, such that the unfiltered signal at time $T+t$ can influence the filtered signal at time T . The value of t depends on the filter specifications, which are given in the methods. When I enter them into the Matlab Filter Designer, I obtain a filter of order 3290, which together with the sampling rate of 25 kHz corresponds to a filter length of 131.6 ms. If this filter is used for a (standard) centered convolution with the data, the value t is half of the filter length, that is $t=65.8$ ms. The main lobes of the filter kernel are contained within 5-10 ms of the center. These values are of relevant size in the context of this study. If the filter is used for a (non-standard) causal convolution, the same applies, because the filter is applied bidirectionally. With bidirectional application, even a IIR filter, which is by construction causal, leads to the described situation. This problem seems to be recognized in the brain-stimulation field, and appropriate modified phase estimation procedures have been developed, see e.g.

Zrenner C, Galevska D, Nieminen JO, Baur D, Stefanou MI, Ziemann U (2020) The shaky ground truth of real-time phase estimation. *NeuroImage* 214:116761.

I am confident that these concerns can be addressed, and I strongly recommend that the authors do so, on principled methodological grounds and also to preempt any criticism by future readers. Notwithstanding these last concerns, this manuscript is of outstanding importance and technical quality. I strongly recommend its publication in *Nature Communications*.

Reviewer #3

(Remarks to the Author)

The revisions are impressive and compelling.

The authors constructively addressed the question of the earlier review and provide remarkable additional findings that support the manuscripts conclusion (new Suppl Fig 8 o frequency specificity, Suppl. Fig 5 on the cycle dependence of the effect, new LFP effect analysis in Fig 3, control of gaze deviation with stimulation in Suppl Fig 1).

This is an elegant, concise, comprehensive and compelling contribution that promises to have high impact.

Version 2:

Reviewer comments:

Reviewer #2

(Remarks to the Author)

The authors have convincingly addressed the one remaining concern. This clarification has strengthened the paper, and I strongly recommend publication and congratulate the authors for a very important contribution.

Dear reviewers,

We appreciated working on your thoughtful questions and queries regarding our manuscript and experiment. We also want to thank you for your time assessing and evaluating our work. The points you made and their inclusion into our work have improved and strengthened the main messages we wanted to convey, of which we are very fond.

In the following, we will include your entire text in red and put our reply to each point in black font below each comment. We have also included the changes in the text resulting from your respective comments below the relevant points.

Thank you again for your invaluable feedback.

Reviewer #1

The elegant manuscript "Gamma-band synchronisation between neurons is causal for effective information processing and behaviour" by Drebitz, Rausch and Kreiter carefully investigates the impact of gamma-phase on the efficacy of an artificial signal to impair a visual perceptual task and evoke spikes in a downstream area. In this study, the authors inserted single pulses of electrical stimulation in extrastriate visual area V2 during a shape tracking task. They observed missed responses and prolonged reaction times dependent on the gamma-phase in V4 representations with overlapping receptive fields. Also, the ability of the artificial stimulation to evoke spikes in V4 was dependent on the V4 phase at the time of the supposed incoming signal.

This study provides important causal evidence for gamma-oscillations as neural mechanism for the effective information processing between visual areas. Data and methods are convincing and well described. There are a couple of areas where some clarifications or additional analyses would be helpful:

1) *Strictly speaking the authors do not show the direct link between the additional spikes generated in V4 and changes in perceptual performance. However, I agree that this is a likely conclusion from the study. I appreciate that the data set and effect might be too small for this direct link, but could the authors report the misses and RT for the trials, for which they could record microstimulation-evoked spikes in V4?*

Following this very interesting suggestion, we took advantage from knowing the effective phase for ICM-evoked afferent input to induce V4 spikes and when these additional ICM-induced V4 spikes occur (as depicted in Fig. 3A in the manuscript). This allowed us to categorize trials based on their V4 spiking activity following ICM pulses that occurred in the last morphing cycle before a behavioral response. We sorted trials in which ICM-evoked afferent input arrived during the effective phase into a group exhibiting more than the average spiking activity in carefully matched periods of trials without ICM and a group in which this was not the case, i.e. with spiking activity comparable to the non-stimulated trials level. The rationale behind this separation aligns precisely with the reviewer's comment: If the additional ICM-evoked spikes in V4 are causally relevant for changes in behavior, we would expect to observe such changes following trials in the first group with enhanced spiking activity (due to ICM).

Indeed, the separation into these groups yielded the predicted result, as shown in Figure 4: Monkeys responded significantly slower in trials where ICM-spikes arrived during the effective γ -phase of V4 activity

and caused additional spikes in V4. This was not the case in the group with the same phase relation but no enhanced spiking in V4.

We incorporated the results of this analysis into the manuscript (around Fig. 4, pages 12 – 13, lines 207-230) and updated the discussion (page 14, lines 245 - 250) materials and methods section (pages: 27 -29, lines 522 -570).

Figure 4

Figure 4: Dependence of RT-delay on ICM-evoked spikes in V4. The two groups represents the differences of RTs between trials with and without ICM-application. The red box represents the difference between RTs of trials, in which the ICM-evoked V2-spikes arrived within the effective V4 γ -phase ($-15^\circ \pm 45^\circ$) and resulted in an above average number of spikes within the following 6 ms (9.2 to 15.2 ms) and those trials without ICM application, but with naturally occurring above average number of spikes within the same time period and following the same γ -phase. The orange box shows the same differences, but for trials and associated RTs in which the ICM-evoked synaptic input did not cause above average spiking activity in V4 and corresponding trials without ICMs that exhibited an average number of spikes. ** indicate significance at $p < 0.01$.

Text results section:

The finding that ICM-evoked input arriving near the peak of the V4 neurons' excitability cycle most effectively perturbs the neuronal spiking activity in V4 and the monkeys' behavior raises the question whether the additional spiking activity directly contributes to the observed behavioral impairment. In support of such a relationship, we found that the delay of RTs caused by ICM-evoked input during the effective phase ($-15^\circ \pm 45^\circ$ at 9.2 ms, see Fig. 3A) depended on the additional spiking activity caused by this input within the response window (9.2 – 15.2 ms after the ICM pulse; Fig. 4). In trials where such ICM pulses caused more than the average number of spikes observed in trials without ICM, the median RT increased by 55.98 ms compared to the median RT of trials without ICM but also an above average number of spikes in the response window (Fig. 4, left box plot; $n = 36$, $p < 0.001$, $z = 3.462$, Wilcoxon rank sum test). In contrast, in trials where these ICM-pulses did not cause an increase in the number of spikes (indicating an unsuccessful interference with stimulus processing despite the same timing of ICM pulses to the effective phase), the median RTs did not change significantly (Fig. 4, right box plot; median -6.66 ms, $n = 105$, $p = 0.252$, $z = 1.146$, Wilcoxon rank sum tests). The RT difference between the two trial groups was highly significant, with a median difference of 62.6 ms ($p < 0.001$, $z = 3.062$, Wilcoxon rank sum test). Importantly, for both monkeys the successfully increased median ICM-evoked spiking activity did not differ significantly from the above average median spiking activity in the non-ICM trials used for comparing RTs (differences in spiking activity monkey B = 6.6 %, $p = 0.146$, monkey T =

8.1% $p = 0.129$, $n_{shuf} = 1000$, bootstrap hypothesis testing). This indicates that the behavioral impairment results from the unpredictable perturbation of the dynamics of cortical processing caused by the ICM-evoked input arriving during the effective phase of the V4 neurons rather than from the higher spike rate evoked by this interference.

Text in Discussion section:

The behavioral impairments following an ICM pulse likely result from the perturbation of the activity pattern in the local population of V4 neurons. This direct link is strongly suggested by the consistent dependence of both effects on the γ -phase of the local V4 population. Moreover, the finding that the extent of RT-prolongation following ICM-evoked synaptic input during the effective γ -phase depends on the magnitude of the resulting additional activity in the V4 neurons provides compelling evidence for this relationship.

Text in Methods section:

Relationship between ICM evoked spiking activity and RTs

To investigate whether there might be a direct link between the additional V4 spikes caused by ICM and the RT delays, we examined how RTs varied depending on the success or failure of ICM pulses in causing additional spikes in V4. To (1) define thresholds for spike detection unperturbed by ICM and (2) to determine the number of spikes expected without ICM pulses, we first configured a surrogate dataset based on the high passed raw signal (same filter settings as in “Methods: Spiking activity”) of trials without ICM and the time points of actually occurring ICM pulses within trials with ICM.

For compiling this surrogate dataset, we used only those ICM pulse times that served to estimate spiking activity after an ICM pulse (Table 1 IIIa). Of these pulses only the last before the behavioral response, and associated with the effective phase range ($-15^\circ \pm 45^\circ$) at 9.2 ms after the ICM were taken into account. Each of these pulse times (template) was then randomly assigned to one of the trials without ICM of the same recording site to position a fictitious ICM pulse. This assignment was constrained to trials without ICM that were sufficiently long to accommodate the respective fictitious ICM pulse and a subsequent period of at least 150 ms before the behavioural response, consistent with the constraints applied to analyses of ICM trials. Within a 10 ms window (centered at 5 ms after the fictitious ICM pulse), we identified the time point with the smallest difference between its γ -phase and the γ -phase at 5 ms after the actual pulse in the template trial. The fictitious ICM pulse was then repositioned to 5 ms before this time point, and the data within the subsequent 6 ms window, starting 9.2 ms after the fictitious pulse, was used later on to estimate the threshold for spike detection and the expected number of spikes within this period. This procedure was repeated 1000 times for each recording site, resulting in the final surrogate dataset.

(1) To define an unperturbed spike detection threshold for each recording site based on the statistical characteristics comparable to those of the signals following the ICM pulses, we calculated the standard deviation of the signal values within each 6 ms surrogate data segment of a given site and averaged them. The recording site’s threshold was then determined based on spike polarity. If the average positive signal components exceeded the absolute average of the negative components, the threshold was set to $+(2 \times (STD)^+)$ and otherwise to $-(2 \times (STD)^-)$.

(2) To determine the number of spikes that are expected to occur in the 6 ms windows from 9.2 ms to 15.2 ms after the ICM pulses, if there would be no ICM pulse, we again used the surrogate dataset based on trials without ICM. We counted how often the signal crossed the recording site-specific threshold within

each of the 6 ms windows following the fictitious pulses for each recording site. This provided a distribution of spike counts for each site, which was used to calculate the expected number of spikes by taking the mean across all spike counts for a given site.

To determine whether, in trials with ICM, an ICM pulse associated with the effective γ -phase range ($-15^\circ \pm 45^\circ$ at 9.2 ms) evoked additional spikes in V4, we first counted the spikes that crossed the recording site-specific threshold within the 6 ms time window between 9.2 ms to 15.2 ms after the pulse and then subtracted the recording site's expected number of spikes for a 6 ms window. The spike counts resulting in differences within the range of $[-1$ to $1]$ were classified as average, while those resulting in differences >1 were classified as above average. The same procedure was performed to classify the spike counts within corresponding 6 ms time windows that follow the fictitious ICM pulses in the surrogate dataset into an above-average and an average group.

To assess changes in RT due to additional V4 spikes that were caused by ICM pulses associated with the effective γ -phase range, we subtracted from the group of RTs following ICM pulses that resulted in above-average spike counts the median of the RTs following fictitious ICM pulses in the recording site's surrogate dataset that resulted in above-average spike counts and compared them with the group of RTs following ICM pulses resulting in average spike counts after subtracting the median of the RTs following fictitious ICM pulses in the recording site's surrogate dataset that resulted in average spike counts.

2.) Also, the authors assert in the first part of the paper that the gamma-phase in V4 matters for the behavioural effect but not the gamma-phase at the stimulation location in V2. Since the authors record also in V2, could they just show the analysis for V2 at the time just prior to stimulation?

We fully agree that investigating the relation between the γ -phase at the V2 stimulation site and the effects of ICM in V2 on V4 neurons and behavior would have been the most straightforward approach to show that the γ -phase of the V2 population is not causal for ICM pulses' effectivity. However, we encountered technical challenges that prevented this analysis.

In the case of monkey T, recordings in area V2 were conducted using the stimulation electrode, which yielded only minimal activity in the γ -band. This outcome is mainly attributable to this electrode's primary function and characteristics optimized for stimulation rather than neuronal recording. Consequently, phase estimates from this noise-dominated signal in the γ -band would necessarily show a random relationship to the behavioral effects. Therefore, the result would be supportive but meaningless.

In monkey B additional V2 sites were recorded, but they were at least 1 mm away from the stimulation site due to the design of their separate microelectrode drives and the recording chamber's grid. Therefore, recorded neurons often differed in the location of their receptive fields, responding e.g. to both stimuli, which rendered them unreliable for estimating γ -phases mimicking those at the stimulation site.

Nevertheless, we identified some sites within monkey B's data that showed responses only to the stimulus within the RF of V2 neurons receiving ICM and with sufficient γ -power. This allowed us to investigate in these additional sites whether the assumption underlying our line of argumentation holds that the pattern of phase coherence between V4 neurons and their afferent inputs from V2 depends on which of the two stimuli in the V4 neurons RF is attended, as it has been observed for V4-V1 pairs (Grothe et al. 2012, Bosman et al. 2012). As expected, we observed a comparable

Phase Coherence V2-V4 for monkey B. The red trace shows the PhC between pairs of V2-V4 sites when both responded to the attended stimulus. The blue trace shows the PhC between the same V2-V4 pairs, but attention is focused on the stimulus outside the V2 RF, but within the V4 RF. The difference between PhC in the γ -band equaled a factor of around 8 (mean over γ -band). The horizontal black lines indicate significance at $p < 0.05$ (bootstrapped hypothesis tests).-

pattern of attention-dependent phase coherence between V2-V4 pairs (see figure below). Importantly, this analysis rests on the limited set of trials without ICMs.

Importantly, neuronal data from these few V2 sites recorded during trials with ICM were unsuitable for analysis due to the large and temporally extended artifacts caused by the electrical currents from the nearby stimulation electrode after each ICM pulse, preventing data analysis around ICM pulses and hence determination of the required γ -phases following ICM pulses.

In summary, technical constraints and limited data prevented the proposed analysis from being as envisaged.

3.)The main behavioural effect is a disturbance of task performance in the form of misses. Therefore, I presume this effect would be less specific with regards to the visual representation than if there were an increase of specific false positives. Apart from the receptive field location, do the authors have any further data on the shape selectivity or the level of stimulus-evoked response of the recorded/stimulated V2 and V4 neurons and how these properties might be related to the probability of a microstimulation effect on task performance?

Our primary observation centered on an increased number of missed responses and prolongation of response times if the ICM-pulse-evoked afferent input arrived during the effective γ -phase in V4. We agree that this effect is rather non-specific and disruptive compared to alterations in shape perception that might increase false positive responses.

However, it is important to note that our experimental setup, especially the high similarity of the shapes, was not made to explore shape tuning and its potential relation to the effects of ICM. This would have required stimulus sets specifically configured for estimating shape-tuning (covering systematically a broader range of differently shaped stimuli with systematic variations but without time-continuous changes of shape) instead of being optimized for high and continuous attentional demands and a large number of presentations. Nevertheless, we tried to assess the shape selectivity of the V4 recording sites.

For monkey T, which has eight potential target shapes, the limited number of trials with only one stimulus in the V4 RF, which we used as a control condition, precluded effective differentiation of neuronal responses to these shapes, especially considering that each of these eight shapes is continuously morphing

from one shape into another shape. In addition, the trial times of these presentations vary, meaning that they could either be during the behaviorally irrelevant MCs 1 and 4 or during the relevant MCs 2 and 3. Thus, the sheer combinatoric variability of presented shape sequences and when and in which behavioral context shapes are presented prevented us from accurately discerning potential differences in response properties.

For monkey B, the conditions for the intended analysis were somewhat better. While the task's stimulus set also contained various shapes (13), only two served as target shapes, thus resulting in a comparatively larger number of presentations of these two shapes. Addressing the problem with the different morphing sequences (different previous and following shapes), we restricted the analysis window to 500 ms centered around the full appearance of each of the two target shapes. However, of the 16 sites analyzed for phase-dependent reaction time (RT) modulation, only 5 exhibited statistically significant differences in response strength to the two target shapes (Wilcoxon rank sum test, $p < 0.05$). The finding of rather small, if any, differences is not unexpected since we have chosen the shapes for their similarity to enhance the need for strong selective attention to the target and not for driving neurons very differently based on their stimulus selectivity.

To see whether there is an indication that differences in response strength might be related to the probability of a microstimulation effect on task performance, we pooled the RTs of trials when the ICM-evoked spikes arrived during the effective phase (as indicated in red in Figure 3) separately for the “preferred” and for the “nonpreferred” stimuli across the few sites with significantly different responses to both target shapes. However, no significant difference in the ICM-dependent increase in RT was observed. The median RT difference between ICM RTs and nonICM RTs for trials in which the shape evoking higher responses (“preferred”) had to be detected was 54 ms ($n = 11$), and the median RT for trials in which the shape evoking lower responses (nonpreferred) had to be detected was 44 ms ($n = 13$, $p = 0.74946$, $z = 0.31935$, Wilcoxon rank sum test). Note that the number of trials, even though they were pooled across the five sites, is so low because we investigated the effect of ICMs on RTs only for those trials in which the ICM evoked spikes arrived during the effective phase of the γ -cycle.

The similar effect sizes suggest that the differential response strength to the target shapes did not translate into observable differences in task performance. This aligns with the assumption of a more general, disturbing effect of the additional input that results in processing disturbance, delaying the response, or even causing a failure to respond. However, the very limited amount of data and the characteristics of the experiment and stimuli certainly preclude any conclusion in this direction based on the recorded neurons' stimulus tuning.

Thus, we refrained from incorporating extensive comments in the manuscript but acknowledged the potential relevance of ICM effects on shape representation. We have included corresponding considerations on these points in the discussion section (pages 15 -16, lines 272 - 288), recognizing their significance in understanding how additional spikes may interfere with processing and perception (see response to the next point for details and the respective text passages).

Text in Discussion section (pages 15 -16, lines 272 - 288):

Whether the ICM-evoked activity modulations in V4 reflect meaningful alterations of shape representations that encode different shapes or correspond to disorganized states that require some additional time to settle again into meaningful states encoding the current stimulus shape remains an open question. The small, essentially random set of afferent axons, each contributing a single additional spike to the local V4

network, the short duration of the activity modulation of only ~5 ms, and the absence of a substantial and consistent increase in false alarms collectively suggest that a meaningful alteration of the perceived shape is less likely.

4.) Such data could provide further insights into the neuronal mechanisms that underlie the perceptual coding in these tasks. The authors speak of the possibilities of evoking phosphenes, which of course would interfere with perception in this task, but could be pretty non-specific to the shape processing. Perhaps the authors would like to speculate more in the discussion about how they envisage the artificial micro-stimulation signal interacts with the shape representations that form the perceptual basis for this task. And perhaps furthermore, what it would take to alter the percept in a meaningful way, now they have an effective mechanism to insert spikes?

We followed these suggestions and incorporated the points raised into our manuscript. The possible role phosphenes is handled now in the discussion section and we extended the discussion about the potential effect of ICMs on information processing in the discussion section (page 14-16, lines 251 -288):

Text in Discussion section:

Several other ways of causing the γ -phase-dependent behavioral impairments are not supported by the results: (1) The ICM-evoked additional spikes did not sufficiently alter activity patterns in area V2 to cause the observed behavioral impairments. If this was the case, ICM applied to the V2-population processing the distracter stimulus should have resulted in much less behavioral impairment than ICM applied to the V2-population processing the target stimulus. Contrary to this expectation, no significant difference was observed between these conditions. Furthermore, the single, low-amplitude ICM pulses applied in V2 likely triggered spikes in only a small number of axons⁴⁵, far fewer than the number of neurons within a V2 column⁴⁶. (2) For similar reasons, ICM-induced phosphenes that might impair the perception of a stimulus's shape are unlikely to explain the behavioral impairments. They are typically induced by pulse trains extending over several hundred milliseconds, with frequencies between 100 and 300 Hz, and not by single pulses^{47–49}. Moreover, impairments by phosphenes should not be independent of their location close to the target or the distracter stimulus. (3) The lack of ICM-induced changes in gaze position rules out the possibility that ICM-induced eye movements explain the observed behavioral effects.

Thus, the brief activity changes during stimulus processing in V4 most likely account for the γ -phase-dependent delays in RTs and the increased rate of missed responses, as the altered activity patterns in V4 momentarily deviate from those associated with normal shape processing. Also, the similarity of the γ -phase dependent prolongation of RT due to ICM across different MCs indicates that the additional input during the effective phase interferes with the fundamental neuronal processing of stimulus information, rather independent of factors like attentional expectancy that change along the trial.

Whether the ICM-evoked activity modulations in V4 reflect meaningful alterations of shape representations that encode different shapes or correspond to disorganized states that require some additional time to settle again into meaningful states encoding the current stimulus shape remains an open question. The small, essentially random set of afferent axons, each contributing a single additional spike to the local V4 network, the short duration of the activity modulation of only ~5 ms, and the absence of a substantial and consistent increase in false alarms collectively suggest that a meaningful alteration of the perceived shape is less likely.

At the end of the discussion section, we discuss the implications of artificial micro-stimulation on altering perception for example by a bidirectional BCI (page 17, lines 309-313):

Text in Discussion section page 17, lines 309-313:

The γ -phase dependence of information routing and processing has practical consequences for future bidirectional brain-computer interfaces. Feeding information into a working brain, as a visual prosthesis would do, requires that the afferent signals arrive during the effective phases of the targeted neurons' γ -oscillatory cycles to ensure effective transmission and joint processing with the target set of brain signals.

5) In the methods under data selection, the authors lay out a number of selection criteria for the different analyses they have conducted. At the moment, the statements are somewhat ambiguous as to whether and at what stage there was some active pre-selection of sessions based on certain expected effects. For instance at the top of page 18, (I) states that for the data shown in Figure 2, only sessions with more misses on ICM than non-ICM trials were included. Then Figure 2A is referenced, which of course includes other sessions, too. Also, it was not entirely clear to me from the results text alone that some of the analyses have been done on a subset of sessions/trials. This is not a problem per se, but should be explicitly stated in the results where applicable.

To improve the description of the data selection criteria, we have included Table 1 in the methods section (page 21), which provides an overview of the selection criteria, including their nature, at which stage and for which analyses they were applied, the number of sessions or sites they were applied to, and how many of them passed the criterion. The detailed description of the selection criteria in the associated text was reworked (pages 22 to 23, lines 394-427).

New text in data selection/methods section:

Data selection

The initial analysis of the effect of ICM on the proportion of missed responses (independent of the γ -phase) is based on all recording sessions in which animals responded correctly in at least 75% of all trials (excluding trials with fixation failures, Table 1: row 1). Recording sessions and their recording sites used for all subsequent analyses fulfilled the following two additional conditions: (I) The proportion of misses in ICM trials had to be higher than in non-ICM trials (sessions fulfilling this criterion fall above the line of identity in Fig. 2A). (II) For each V4 recording site the average γ -band power (52–118 Hz, see Drebitz et al. 20186) during MCs 2 and 3 across successful trials had to be at least 70 % higher than that during the 800 ms interval beginning 70 ms after baseline period onset, Fig. 1C) in each of the two task conditions with a single, attended stimulus present within the V4 RF.

The data used for all subsequently listed behavioural and neurophysiological analyses came from successful trials (except when misses were required) of the two task conditions requiring selective attention for one of the two stimuli in the V4 RF. (IIIa) To examine the γ -phase dependence of the effect of ICM on the rate of misses and response times (RTs) and to analyse the relation between ICM-evoked spikes and RTs, we excluded recording sites outside the granular layer of area V4. This was necessary to prevent averaging across sites in different layers, as γ -oscillatory activities are laminar specific and exhibit characteristic phase shifts across layers 55,66.

(IIIb) To analyze the effect of ICM on V4 neurons' spiking activity and the LFP depending on the phase of the local γ -oscillatory activity, we required a significant enhancement in the spiking activity in response to visual stimulation during MCs 2 and 3 compared to the spiking activity in the 800 ms interval beginning 70

ms after baseline period onset in both conditions with a single attended stimulus within the V4 RF (Wilcoxon rank sum test, $p \leq 0.05$).

Trials satisfying the above criteria provided up to four 650 ms segments of neuronal data from the V4 sites, which starting 500 ms before each ICM pulse. To be included, each had to fulfill the following additional criteria: (1) More than 150 ms had to pass between the ICM pulse and the behavioral response. (2) To ensure a sufficiently large V4 γ -oscillation amplitude for a reliable computation of the γ -phase 5 ms after the ICM pulse in the stimulated trials (and 5 ms after the fictitious ICM pulse in trials without ICM used for generating surrogates; see Statistical evaluation), we required at that time a γ -power $P(t)$ of the LFP signal larger than the median of the 650 γ -power values along the segment. (3) To analyze the γ -phase-dependence of misses and RTs, data segments (and corresponding ICM pulses) had to come from the terminal MC of trials where animals either failed to respond or responded correctly. (4) To analyze the spiking activity, all (up to four) data segments around ICM pulses within a trial were used (provided that they fulfilled criteria (1) and (2)).

The reference in the methods section to Figure 2A was intended to indicate that not all sessions fulfilled criterion (I), as evident in the scatter plots. To further aid clarity, we have added an explanation in brackets, guiding readers on what to observe in Fig. 2A (page 22, lines 398-399:

"(sessions fulfilling this criterion fall above the line of identity in Fig. 2A).

It would also be helpful to the reader to give an indication of the number of excluded sessions at each stage of the analysis, starting from the initially recorded sessions. This could be done in the results in the text or perhaps as a table in the methods, referencing each Figure/Analysis.

We have addressed this suggestion by including this information into Table 1 in the Methods section, as described in the previous point.

Reviewer #2

1) Fig. 2B, C: I assume that the x-axes show the V4 LFP gamma phase after shifting it according to the V4 ESA, such that phase zero has maximal ESA. If so, the fact that the behavioral data, i.e. the y-axis values, peak at zero is a finding. Alternatively, the y-axis values have been realigned such that their peak is at zero. The legend states "The curve's maximum is centered at 0°", and it is not clear to this reviewer, whether this describes a finding or an operation performed by the authors during data analysis. If the authors centered the curve's maxima at zero, it would be trivial that the values decline for other x-axis values (this would be particularly concerning, if the centering would have been done per session, before averaging over sessions). Please clarify.

The order of steps to arrive at the figures is the following: First, the phase of the LFP's γ -band oscillations was corrected (shifted) according to the location of the ESA-peak (spiking activity) for each recording site individually. This correction ensured that the highest spiking activity in the LFP's γ -cycle coincided with its trough, corresponding to the peak of the neurons' excitability cycle at 0° . Second, for each recording site, the RTs and missed responses were sorted according to the associated γ -phase value of the V4 neurons' excitability cycle 5 ms after the last ICM before the behavioral response (or miss). This sorting resulted for each recording site in a curve describing the phase-dependent distribution of misses and response times, as explained in the methods section (Phase dependence of the missed responses, RTs, LFP, and ESA; pages 25-27, lines 474-521). Third, we averaged across all these curves of both animals combined (Figure 2B) or separately (Figure 2C). Therefore, the pronounced phase-dependent modulation in the resulting average curves is not a trivial result of centering the maxima of the individual site-specific curves at 0° before averaging (since no such operation was performed for the individual site-specific curves). However, since the phase values were taken at a point in time that was selected for methodological reasons and could have been different, the specific phase values of these curves were, at that stage of the analyses, essentially meaningless and hardly interpretable for the reader. Therefore, we shifted the maxima of these final average curves to 0° , thereby improving the visibility of the curves' unimodal shape and preventing readers from interpreting these phase values. The calculation of meaningful and interpretable phase values becomes possible only at a later stage of the analysis in the results section when the time at which the ICM-evoked input arrives has been derived, and phases at this point in time can be calculated (around Figure 3, page 10).

To clarify this issue and avoid confusion, we have changed the misleading statements in the legend of Figures 2B and 2C, which now reads:

Text in caption Figure 2B/C:

We centered the median curve's maximum at 0° for better visibility of the unimodal shape.

2.) Line 360: "The regions $\pm 45^\circ$ around the maximum and minimum of ~~eeeeeead~~($\varphi\varphi$) or ~~mmmmmm~~($\varphi\varphi$) define the effective (EPR) and ineffective phase range (IPR), respectively, for statistical analyses". By doing it like this, the authors essentially show merely that the dependent variable shows some dependence on the independent variable. In principle, for their statistics to become significant, the curves shown e.g. in Fig. 2B,C could take any shape, as long as the maximum and minimum have sufficiently different y-values; the maximum and minimum could e.g. occur at very nearby x-values, and the curve could be very different from a cosine. By contrast, the curves shown e.g. in Fig. 2B,C could probably be fitted well by a cosine, or a von-Mises function. I suggest that the authors consider using this observation, because it makes for a much stronger case. Also, some readers might consider it

problematic if the authors first select maxima and minima, and then show that they are different. I think that the observed differences are not trivial, but I also think that the results support stronger inferences. At the least, the authors could simply fix the EPR to be centered at zero, and the IPR to be centered at 180 degrees, i.e. to be defined by the ESA. This would avoid concerns about circularity (assuming that zero phase is defined solely on the basis of the ESA – see my previous comment).

Regarding the suggestion to consider fitting the phase-dependent curves with cosine or von Mises functions, we acknowledge the potential for strengthening our analysis by employing such approaches. However, it is important to note that our study investigates whether the phase of neurons' γ -excitability cycle is causal for the effectiveness of afferent input. The description in the previous manuscript might have been misleading (and was adjusted based on the comments to the previous point), as we did not clearly explain that we use the γ -LFP to obtain phase values but finally investigate the γ -excitability, which is clearly related but not necessarily identically shaped. Thus, even though the suggested symmetric functions are likely capable of explaining the shape of the LFP, this might not necessarily hold for the excitability cycle.

Given the non-linearities of, for example, spike generation, the subsequent refractory periods, as well as the arrival of inhibitory inputs, which are commonly assumed to be required for the generation of γ -oscillatory excitability and are assumed to arrive around 90° after the peak of the excitatory cycle (see for example: Börgers and Kopell (2003) "*Synchronization in Networks of Excitatory and Inhibitory Neurons with Sparse, Random Connectivity*" or Brunel and Wang (2003) "*What Determines the Frequency of Fast Network Oscillations With Irregular Neural Discharges? I. Synaptic Dynamics and Excitation-Inhibition Balance*", which both suggest that inhibitory input arrives around 90° and not at 180° after the peak of excitation suggesting an asymmetric excitability cycle; for review: Tiesinga and Sejnowski (2009) "*Cortical Enlightenment: Are Attentional Gamma Oscillations Driven by ING or PING?*"). Consequently an asymmetry of the γ -excitability cycle is certainly a plausible possibility. Thus, we are cautious about assuming a sinusoidal or otherwise symmetric modulation of the γ -excitability dependent effects without sufficient evidence to support such an assumption.

Concerning the potential impression of circularity in defining maxima and minima and subsequently demonstrating their differences, we conducted shuffle controls to assess the probability of observing by chance the peak-to-peak amplitudes of our modulation curves (maximum-minimum of missed responses and RTs). For this purpose, we randomized 10,000 times the relationship between RTs (or misses and correct responses) and γ -phases by randomly reassigning RTs (or correct responses and misses) to γ -phases. We recalculated the modulation curves and their peak-to-peak amplitudes. This analysis revealed that our observed modulation was significantly larger than expected by chance. We added these results at the appropriate passages in the manuscript.

Results section page 7 lines 100-103:

The difference in the number misses between the most effective and the ineffective phase ranges was significantly larger than expected by chance (nshuf = 10,000, $p = 0.0021$, bootstrap hypothesis testing).

and on page 8 lines 115-118:

For both animals, the modulation of RTs between effective and ineffective phase ranges was significantly larger than expected by chance (upper panel monkey B: nshuf=10,000, $p = 0.0004$, lower panel monkey T: nshuf = 10,000, $p = 0.0005$, both bootstrap hypothesis testing).

The Methods section was extended correspondingly pages 33-34, lines 662-670:

To test whether the modulation depth of response times and misses as a function of the γ -phase was statistically significant (Fig. 2 B/C), we performed shuffle controls. This procedure randomly reassigned the RTs and the correct responses and misses to the γ -phases for each recording site. Then, the modulation curves were obtained as described in the methods section: "Phase dependence of the missed responses, RTs, LFP, and ESA", and the curves' peak-to-peak amplitudes (difference between minimum and maximum) were calculated. This procedure was repeated $n_{\text{shuf}} = 10,000$ times to generate a distribution of values representing the chance level. This distribution was used to estimate the probability that the observed peak-to-peak amplitudes of RTs and missed responses following ICMs were significantly different from chance.

3) Similar concerns apply to the analysis that first selects the maximum ESA response and then compares it to a surrogate distribution (line 150). Some readers might perceive this as circular. I think it is not circular, it is just an analysis that is less strong than it could probably be, given the presented results. I find more elegant the analysis presented in lines 156-159, which avoids the selection of the maximum.

We agree that the analysis is not circular, because we perform the identification of maxima in the surrogate data exactly as we do in the ICM data. All statistical characteristics are kept equal, the only difference is whether ICM was applied or not. We hope to avoid the perception of circularity by changing the wording in the text and adding "corresponding" (page 11, line 174), to indicate that the surrogate maxima were identified as has been done in the ICM data.

We certainly need to report the size of the peak and therefore it appears meaningful to test whether it is in a range of values one could expect even in data without ICM application, or whether it differs significantly from it.

4.) The authors use three different delays after the ICM pulse for their analyses:

Line 144: "These gamma phase were estimated at 10 ms after the ICM pulse"

Line 336: "To analyze the phase dependence, we used the γ -phase $\Phi_{\text{LFP}}(t)$ with t_{delay} set to 5 ms after the ICM pulse."

Line 346: "shared γ -phase at 9.2 ms after the ICM pulse that fell into the same 10° phase bin". How were the values of 5 ms, 9.2 ms, and 10 ms chosen? Why were there three different values? Also, I do not understand what is meant by "that fell into the same 10° phase bin"; please clarify. These questions are partially answered around line 170, but the authors should explain this earlier and in more detail.

We apologize for the confusion regarding the γ -phase estimation. To address this, we have first clarified in the Methods section that phase values refer in general to the phase of the excitability cycle $\Phi_{\text{Exc}}(t)$ and explain how the excitability cycle of the neurons differs from the oscillatory cycle of the LFP and how its phase values are derived from the phase values for the LFP $\Phi_{\text{LFP}}(t)$. To enhance clarity, we now state in the methods section that the phase values at 10 ms and 9.2 ms delays are values derived from the excitability cycle's γ -phase at 5 ms (pages 24-25 lines 455-472):

Text in methods section pages 24-25 lines 455-472):

To correct this and express the phase values in terms of the excitability cycle, we shifted the γ -phases derived from the LFP such that within the average γ -cycle, the maximum ESA occurred at 0° (see also Lisitsyn et al.52). To this end, we used the ESA-signal and LFP γ -phase values during MCs 2 and 3 of all trials with and without ICM, except for the periods between 20 ms before and 50 ms after an ICM pulse to compute the means of the ESA values sorted into 15° bins according to the γ -phase of the LFP for each V4 recording site and fitted the resulting histogram with the function $y(\varphi)=a * [\cos(\omega(\varphi+b))] ^ c$ (with $a>=0$, $-\pi <= b < \pi$, and $c >= 0$). Then, the phase shift b was added to the recording site's phase values of the LFP to obtain $\varphi_ExC(t)$, the γ -phase of the excitability cycle at times t .

To analyze the phase dependence of the behavioral and physiological consequences of ICM pulses, we used the excitability cycle's γ -phase $\varphi_ExC(t)$ with t set to 5 ms after the ICM pulse to obtain reliable phase values unperturbed by ICM-evoked activity in V4 or potential remnants of the electrical artifact. The γ -phases at later time points of interest (10 and 9.2 ms) were calculated based on $\varphi_ExC(t)$ at 5 ms, the time difference between 5ms and 9.2 or 10 ms, and the animals' average γ -peak-frequency (SFig. 1, SNotes 1). The latter was calculated based on the mean phase progression of $\varphi_ExC(t)$ between 5 ms before and 5 ms after the ICM pulse across all selected data segments for each animal.

The reasons why three different delays after the ICM pulse occur in the text are the following:

The delay of 5 ms was chosen since any potential remnants of the electrical artifact have decayed at this point while still, no ICM-evoked activity from V2 has arrived. This allows for the obtaining of reliable, unperturbed phase values. At the same time, this point is close enough to physiologically relevant points around 9-10 ms after the ICM pulse when ICM-evoked spikes from V2 arrive and interact with V4 neurons. This allows using a forecasting approach based on the average individual γ -frequency to calculate with sufficient precision the phase values around 9-10 ms, which cannot be taken directly from the signal that might already be influenced by an evoked response. Thus, the 5 ms delay (mentioned only in the methods section) is only a methodologically necessary delay at which reliable phase values were taken solely to calculate the phase values at the relevant points around 9 to 10 ms.

The two other delays after the ICM pulse that show up in the reported results were used for the following reasons:

The analysis of the time course of V4 activity following an ICM pulse, depending on the phase when the evoked afferent activity arrives (as shown in the heatmaps of Figure 3), requires selecting a specific point in time to estimate the phase, which serves to group and average the snippets of ESA and LFP signals for the analysis. While in principle, this could have been achieved with the phase values taken at 5 ms, using this early time point would have resulted in phase values (shown on the γ -axis of the heatmaps) that are not well interpretable, as they do not correspond to the phases at the actual time when ICM-evoked afferent input arrives. Therefore,

we opted for a later time, closer to the latency for the arrival of the afferent input from V2 in V4, which is around 10 ms. Then the results of this analysis allowed us to estimate better this latency, which we found to be 9.2 ms, as we explain in line 199, page 12.

We also revised the wording of the second sentence cited from line 144 (now line 166-167), which likely promoted the impression that phase values were taken directly from the LFP:

These γ -phases were calculated for the time 10 ms after the ICM pulse, which is close to the time when the ICM-evoked spikes arrived in V4 (see Materials and Methods for details)." (page 11, lines 166-167).

Concerning the comment on the lack of clarity in the phrase "that fell into the same 10° phase bin" we apologize for the erroneous sentence. We essentially constructed for the trials of each recording site a histogram that finally provided the percentage of misses among correct and missed responses depending on the γ -phase at 9.2 ms after the ICM-pulse. Since the bin width of this histogram was 10°, there are groups of trials with γ -phase values that belong to the same 10° phase bin (therefore, the formulation "that fell into the same 10° phase bin"). We modified the text to remove the erroneous sentence and enhance the clarity of the description.

Text in methods section page 25 lines 476-480

For each selected recording site in both animals, we constructed a histogram providing the number of misses across γ -phase at 9.2 ms after the ICM pulse with bins of 10° width. Each bin's count was normalized by dividing with the number of trials with γ -phase values in the bin's phase range that resulted in correct responses or misses.

Further points:

Line 35: "Essentially, neurons respond as if only the attended stimulus is present". I suggest to specify that this holds for neurons in higher areas, with two competing stimuli in their RF.

We've incorporated the suggestion by specifying that neurons with multiple stimuli in their receptive fields respond as if only the attended stimulus is present (page 2. Lines 34-38):

In the visual cortex, selective attention resolves this competition in favor of the most behaviorally relevant stimulus: Essentially, neurons with multiple stimuli in their RFs respond as if only the attended stimulus is present by selectively processing the signals obtained by that subset of their afferent inputs, which provides information on the attended stimulus while suppressing other signals.

In Fig. 1B, the blue RF and labeling seems to have a low contrast relative to the grey background.

We changed the figure accordingly.

In the legend for Fig. 2B, I found this description confusing: "The graph shows the median proportion of MRs between correct and missed responses". The methods states: "percentage of trials resulting in missed responses among all trials that resulted in correct or missed responses", which is much clearer.

We changed the corresponding description in the legend of Figure 2 accordingly.

Line 228: I presume that the cue, i.e. the annulus, disappeared before the shapes appeared. Please clarify.

Yes, the cue disappeared when the lever was pressed. We included a statement explicitly stating this in lines 331-335, on page 18

The animals had to fixate on the FP and press a lever to start the trial, upon which the cue vanished. After a baseline period of 1050 ms with no stimulus on the monitor but the FP, three or four stimuli, each with a complex-shaped contour ($\sim 1.5^\circ$ diameter, line width 0.25° , 3.8 cd/m^2 , Fig. 1B/C), appeared at isoeccentric locations on the screen.

Line 330 refers to "Lisitsyn et al., 2020", whereas the referencing style and the bibliography are numbered.

We name and refer to this and a few other references by naming the authors for stylistic reasons, which we restricted to the methods section. We checked if this is in accordance with the reference style of Nature papers and found several examples in published articles in Nature Communications, which did so. However, in most cases, authors did not include the year of publication (for example: Yiling et al., 2023: "Robust encoding of natural stimuli by neuronal response sequences in monkey visual cortex"). We did update the few cases in which we refer to a reference with the author names by stating the first author's name and adding the corresponding number in the bibliography to ensure consistency while not mentioning the year of publication.

The authors shift the LFP phase estimates to align maximal ESA to phase zero. To clearly label this shifted LFP phase as such, they might consider to refer to it throughout as "corrected", "shifted", or "aligned" phase, or similar, and also use a respective abbreviation for it.

In part we dealt with this issue already at the previous request (starting with: "The authors use three different delays after the ICM pulse for their analyses: ..."). We do shift the LFP phases based on each recording site's γ -oscillatory activity cycle (measured as ESA across the γ -oscillatory LFP's phase), and we refer to the now shifted, γ -phase values as the phase of the γ -excitability cycle (ExC) throughout the manuscript. However, this was explained a bit misleadingly in the first version of the methods section, which was changed to increase clarity. Throughout the manuscript, we refer to the γ -phases of this "corrected, shifted or aligned" rhythmic activity based on the recorded LFP as phases of the γ -excitability cycle of the recorded V4 neurons. This terminology also expresses the meaning of the measure and should therefore help readers that are less familiar with the field.

Change in methods section (page 24-25, lines 455-472):

To correct this and express the phase values in terms of the excitability cycle, we shifted the γ -phases derived from the LFP such that within the average γ -cycle, the maximum ESA occurred at 0° (see also Lisitsyn et al). To this end, we used the ESA-signal and LFP γ -phase values during MCs 2 and 3 of all trials with and without ICM, except for the periods between 20 ms before and 50 ms after an ICM pulse to compute the means of the ESA values sorted into 15° bins according to the γ -phase of the LFP for each V4 recording site and fitted the resulting histogram with the function

$y(\varphi) = a * [\cos(\varphi + b)] + c$ (with $a \geq 0$, $-\pi \leq b < \pi$, and $c \geq 0$). Then, the phase shift b was added to the recording site's phase values of the LFP to obtain $ExC(t)$, the γ -phase of the excitability cycle at times t .

To analyze the phase dependence of the behavioral and physiological consequences of ICM pulses, we used the excitability cycle's γ -phase $ExC(t)$ with t set to 5 ms after the ICM pulse to obtain reliable phase values unperturbed by ICM-evoked activity in V4 or potential remnants of the electrical artifact. The γ -phases at later time points of interest (10 and 9.2 ms) were calculated based on $ExC(t)$ at 5 ms, the time difference between 5 ms and 9.2 or 10 ms, and the animals' average γ -peak-frequency (SFig. 1, SNotes 1). The latter was calculated based on the mean phase progression of $ExC(t)$ between 5 ms before and 5 ms after the ICM pulse across all selected data segments for each animal.

Further changes in tables page 26 lines 488 and 498

And Page 26 line 500:

..... [where n is the number of all selected data segments, $ESA_{(i,t)}$ is the ESA value at time t of data segment i , $ExC(t)$ is the γ -phase of the excitability cycle at time t in data segment i]....

Fig. 2B,C: I guess that B shows the pooled data of both monkeys, and C shows the individual data per monkey. Please clarify in the figure legend or by respective labeling of the figure panels.

We changed the text in the legend accordingly.

Line 88 and Stable 1: It seems that ICM reduced the false alarm rate in monkey T. This should be mentioned and discussed.

As suggested, we now mention this observation for monkey T in **lines 94-97 (page 7)**:

The ICM pulses did not significantly affect the number of fixation failures, while the percentage of false alarms decreased significantly for monkey T, but not for monkey B (STable 1). These findings indicate that the volley of spikes evoked by a single ICM pulse can effectively interfere with information processing and, ultimately, with behavior.

However, there are multiple potential explanations for this observation. Especially considering the differing outcomes between monkey T and monkey B, which makes our explanations somewhat speculative. For monkey T, the reduction in false alarms could be attributed to several factors. One plausible explanation is a shift in decision criterion in response to the increased difficulty of ICM trials. It is conceivable that the higher difficulty prompted monkey T to demand more evidence before perceiving a target shape, leading to a higher response criterion during ICM trials than non-ICM trials. In contrast, monkey B may not exhibit a similar adjustment in decision criterion.

Another potential explanation might be related to the differences in tasks between animals. Monkey T learned to respond to more initial shapes (serving at the cued location as a sample) than monkey B did,

and these shapes also appeared as non-matching test shapes (if not selected as initial shape) within the morphing sequence of the cued stimulus throughout a trial. For monkey B, who learned fewer potential initial shapes (serving as samples), the likelihood of such a shape appearing as non-matching test shapes in the sequence of a cued stimulus was much smaller. If the ICM pulse somehow perturbs the perception of these stimuli to which the monkeys erroneously tend to respond, then in monkey T there are more occasions when false alarms are prevented by degrading perception of the more often appearing shapes that are most likely to trigger this behavioral error.

One could also speculate that such changes in performance could be caused by (unlikely) phosphenes or other ICM-induced phenomena that perturb the representation of the actual shape .

However, the differences in results between the two monkeys make it hard to draw conclusions on the underlying reasons. Given the multitude of speculative explanations, we believe it is prudent to refrain from extensive discussion in the manuscript.

Line 124: This control is extremely elegant and convincing. Can the authors show for the present data the effect of attention on V2-V4 coherence, which is described in lines 120-123?

Unfortunately, due to several aspects of the experimental design being optimized to investigate the phase-dependent consequences of ICM pulses and not attention-dependent phase coherence, a proper analysis of attention-dependent phase coherence was not possible:

- First, for both animals, using the much large number of trials with ICM pulses was impossible since the close distance of the V2 electrodes to the stimulation electrode (and of course the stimulation electrode itself) resulted in the superimposition of their data with large and long-lasting stimulation artifacts that could not be removed.
- Second, for both animals, the characteristics of the stimulation electrode were selected such that it was optimized for delivering ICM pulses, which on the other hand was not optimal for recording neuronal data. These electrodes delivered only poor oscillatory signals, which is attributable to this electrode's primary function and characteristics optimized for stimulation. Consequently, phase estimates from this noise-dominated signal in the γ -band would necessarily show a random relationship to the behavioral effects. Therefore, the result would be meaningless. For monkey T, the stimulation electrode was the only potential source for V2 data.
- For monkey B, despite recording from additional V2 sites with RFs that partially overlapped with the RF of the stimulated site, these sites often responded to both stimuli due to spatial constraints in the recording chamber. Consequently, data from these sites could not be used for the suggested analysis, leaving only a limited number of pairs of V2-V4 sites ($n=34$) that met the inclusion criteria. However, each of these sites provided only a few trials without ICM required for the suggested analysis.

Pooling these few trials across V2-V4 pairs in monkey B allowed for calculating the PhC within MCs 2 and 3 and both stimuli within the V4 RF for attention directed either to the stimulus (partially) covering the RF of the V2 recording site (attend in V2 RF) or to the other stimulus within the V4 RF that did not drive the recorded V2 site (attend out V2 RF). The figure above shows a significant difference in PhC during the two task conditions (attend in V2 RF: red line; attended out RF2 blue line). As expected, the difference was highly significant ($p = 0.0096933$, paired t-test for values at 63 Hz bin), with phase coherence almost vanishing when the stimulus activating the V2 site was not attended.

However, since the V2 data of monkey B were also far from optimal due to the experiment not being designed for this type of analysis, we believe it is prudent to refrain from including such an ad hoc analysis of phase coherence between V2 and V4 in our manuscript.

Phase Coherence V2-V4 for monkey B. The red trace shows the PhC between pairs of V2-V4 sites when both responded to the attended stimulus. The blue trace shows the PhC between the same V2-V4 pairs, but attention is focused on the stimulus outside the V2 RF, but within the V4 RF. The difference between PhC in the γ -band equaled a factor of around 8 (mean over γ -band). The horizontal black lines indicate significance at $p < 0.05$ (bootstrapped hypothesis tests).-

Reviewer #3

1.) The interpretation of the authors point to the gamma phase as the key determinant of the evoked spiking and behavioral effect. It would clarify the frequency specificity if the authors could identify a frequency range (towards low beta or higher gamma) at which the phase dependency of the effect disappears. This could also help future studies when identifying the frequency at which phase estimates can meaningfully predict stimulation effects.

To address the question in which frequency ranges the observed phase-dependent increase of RTs disappears, we sorted all periods around ICM-pulses directly preceding the behavioral response according to the dominant frequency of the V4 excitatory activity and compared the effect size across different frequencies.

In more detail:

- We first filtered the raw signals with a broad bandpass filter with pass frequencies between 17 and 120 Hz, encompassing the spectrum's beta and the high gamma regions.
- Then, the dominant frequency around the time of ICM was calculated based on the average phase progression in the period -5 to 15 ms around the stimulation pulse. This period is centered at 5 ms after the ICM, the point in time used for phase estimations. The phase values were derived from the Hilbert transform of the band passed (17-120 Hz) signal, and then the phase progression within the 20 ms window was calculated. The instantaneous frequency was calculated based on this average phase progression in 20 ms. The figure below shows the distribution of instantaneous frequencies actually observed around the time of ICM pulses directly preceding the behavioral response
- The frequencies within a rather broad γ -frequency range (roughly 40 – 80 Hz) were most commonly observed for both animals. Instantaneous frequencies in the direction of the β - or high γ -band were rarely observed.

- Next, we calculated the potential phase-dependent effects of ICM pulses based on the data within 20 Hz wide frequency windows positioned in 5 Hz steps across the frequency range with sufficient data to calculate the γ -phase dependent RT modulation curves for the individual sites. This allowed us to start with a frequency window between 25 to 45 Hz and to continue up a window between 80 and 100 Hz.
- For each of the 20 Hz wide frequency windows, we first computed a median RT-modulation curve across all sites as a function of the γ -phase at 5 ms after the ICM pulse for each animal individually (as it was done for the analysis in Figure 2C). Based on these curves, the effective and ineffective phase ranges for each frequency window of each animal were defined as the phase range of $\pm 45^\circ$ centered around the maximum and minimum of the median modulation curve, respectively. Subsequently, the RTs associated with either the effective or ineffective phase range were pooled across the recording sites of both animals. This pooling was necessary since the number of RTs drastically decreases when considering only cases within a 20 Hz frequency window compared to the much wider frequency band used for the data shown in Figure 2C.
- To assess the strength of the γ -phase dependent effect of ICM pulses on RTs, we calculated the difference between the median RTs of the effective phase and the median RTs falling into the ineffective phase for each frequency range (Figure above, black trace with dots).
- We performed shuffle controls to assess whether the observed difference between RTs associated with the ineffective and the effective γ -phases was larger than chance. Therefore, for each site and each frequency range (20 Hz window), the RTs were randomly

reassigned to γ -phases and instantaneous frequencies, and the frequency-window-dependent differences of RTs were calculated as just described, but for these shuffled data. This procedure was repeated 5000 times. The figure above shows the median spread between RTs falling into the “random” maxima and minima across the 5000 individual interactions as a dashed line. The gray shading indicates the first and third quantiles of the shuffle controls. The horizontal black bars indicate the effects of the ICMs on RTs with likelihoods $p < 0.05$.

- We found that the significant γ -phase-dependent effect of ICM was evident for frequency ranges of the γ -band (45-80 Hz center frequencies). At the same time, we observed no effect in the direction of the beta or high gamma band, where we observed only a small number of cases with these frequencies anyway. –

We included the analysis of the frequency dependence of our observed effect into the supplementary information (SFig. 8) page 12, and included a reference to it in the manuscript **page 8, lines 133-136:**

We also tested whether the γ -phase-dependent effect of ICM on RTs could be attributed to specific frequencies within a broad γ -frequency range. We observed similar effect sizes for frequencies between 40 and 80 Hz, while the effect vanished toward the higher γ - and β -frequency ranges (SFig. 7).

2.) The microsimulation was delivered during the 1st, 2nd, 3rd, or 4th morphing cycle. Can the authors add information clarifying that the relative time of the stimulation pulse does not change the main effect,, i.e. that the phase-dependent evoked field amplitude (or the behavioral misses) is qualitatively similar for early and later stimulation pulses. A difference of the evoked field would not question the papers key results, but it could help interpreting the effect as being linked or not linked to attentional expectancy.

- To investigate this issue, we analyzed the delay induced by ICMs at the end of morphing cycles 2, 3, and 4. We lack RT data at the end of morph cycle 1 due to the task design, as shapes must morph away from the initial shape before reappearing at the end of cycle 2 or later. We used RTs instead of the number of misses since the combined sample size for RTs of both animals allows for splitting into groups for cycles 2, 3, and 4. In contrast, the sample size for the comparatively rare misses was insufficient for this splitting, even when pooling across animals.

Left figure: Delay of RTs during the effective phase ranges of both animals for morphing cycles two, three and four, as difference to the median RTs of trials without ICM-application (for each MC separately). Due to the separation into three groups, we excluded days with less than 5 entries during the effective phase range and pooled data of both animals. The median values for MC2 (40.75 ms) and MC3 (41.25 ms) were not significantly different even before Bonferroni correction ($p = 0.14497$), while the differences between MC2 and MC4 (-21.78 ms) survived the multicomparison corrections ($p = 0.00042916$), as well as the differences between RTs of MCs 3 and MCs 4 ($p = 0.0063333$, all comparisons Wilcoxon signed-rank tests, significant p -values are Bonferroni-corrected)

- Our analysis revealed no significant differences in the delay induced by ICMs for the effective phase ranges between cycles 2 and 3 (see Supplementary Figure 5). However, we did observe differences in the effect between cycles 2 and 3 compared to cycle 4. Interestingly, the effect essentially disappeared for ICMs applied during cycle 4. For MC 4 we observed no significant difference between RTs observed following ICMs associated with the effective γ -phase and RTs derived from trials without ICMs of the same MC ($p = 0.075429$, $z = 1.7778$, Wilcoxon signed-rank test). Also, the RTs following ICMs associated with the effective and the ineffective group (not shown here) did not differ significantly for MC 4 (median effective: -21.78 ms, median ineffective: -14.515, $p = 0.18237$, $z = 1.3335$, Wilcoxon signed-rank test), while these two groups differed significantly for RTs at the end of MC 2 and MC 3 respectively (see SFig 5). We did not analyze RTs during cycle 4 from the beginning on because the animals do not need to actively process and evaluate the shape during this phase, as they learned (through training) that the shape will appear at the end of cycle 4 with 100% likelihood. Consequently, they likely time the release of the lever in this last cycle instead of selectively following the shape during the morphing process. This also explains the rare occurrence of misses at the end of cycle 4.
- We have included the separate analysis of RTs for cycles 2 and 3 (since we think MC 4 does not deliver information about the γ -phase dependent impact of ICMs) and the corresponding comparisons in the supplementary information (page 10, SFig. 5) to demonstrate that ICMs are similarly effective during both periods of the trial. Additionally,

we have added a note **in line 122-126 (page 8)** of the main text to report that the effect was consistent across cycles 2 and 3:

In addition, the differences between RTs for both phase ranges were highly significant (monkey B: 45.0 ms, n = 16, p = 0.002, t = 3.3579; monkey T: 48.9 ms, n = 18, p = 0.012, t = 2.657; paired Student's t-tests) and this effect was equally strong for RTs following the reappearance of the initial shape at the end of MCs 2 and 3 (SFig. 5).

And in the discussion section (page 15 lines 267-271):

Also, the similarity of the γ -phase dependent prolongation of RT due to ICM across different MCs indicates that the additional input during the effective phase interferes with the fundamental neuronal processing of stimulus information, rather independent of factors like attentional expectancy that change along the trial.

3.) It is difficult to discern for a reader whether the stimulation effect on V4 responses varied between electrode contacts from different layers of the V4 site. Is it possible to distinguish layers, or upper versus lower recording sites, to constrain the interpretation of the findings to be potentially stronger for the upper V4 layers ?

We indeed classified the approximate location of our V4 electrodes, primarily targeting upper layers (granular and supragranular layers) during recordings. Out of 119 recording sites, 34 were not assigned to a layer due to unusual onset patterns, and only 5 recording sites were located in infragranular layers (too few for a subset analysis). While considering separating the data into subsets of upper locations, we encountered challenges due to the disparate distribution of sites across animals. Monkey B had 17 granular and 32 supragranular sites, while Monkey T had 24 granular and 7 supragranular sites. This would have led to an imbalance in subset sizes, with the supragranular subset primarily comprising data from Monkey B. Consequently, any observed differences between layers could be influenced by both location and animal composition. Hence, we refrain from making conclusions regarding potential laminar differences based on our dataset. To clarify the selection of electrodes based on their laminar location in our manuscript, we updated the methods section "data selection" The text now explains for which analysis the laminar location was relevant (Table 1) and why (**page22, lines 406-410**):

(IIIa) To examine the γ -phase dependence of the effect of ICM on the rate of misses and response times (RTs) and to analyse the relation between ICM-evoked spikes and RTs, we excluded recording sites outside the granular layer of area V4. This was necessary to prevent averaging across sites in different layers, as γ -oscillatory activities are laminar specific and exhibit characteristic phase shifts across layers.

We decided to restrict our analysis of γ -phase dependent effects of ICMs on behaviour to granular sites, as this subset of sites contained a similar number of recordings from both animals, and we wanted to avoid averaging sites from different layers that are known to be shifted in phase.

We did not restrict the analysis of spiking activity to granular sites because we could individually align the ongoing oscillatory activity to the spiking activity (the trough of the gamma LFP to the peak spiking activity) for each site, thereby correcting for potential phase differences across layers.

4.) It is explicitly acknowledged that the authors are focusing on the (entire average) spiking responses as their metric evaluating stimulation effects, which is an exceptional success. I can imagine that the readership, however, will include human noninvasive EEG researchers, who will wonder whether the evoked stimulation effects are not also, or even stronger and longer, showing up in the local field potential measure. Describing whether there was any hint of an evoked field effect could enhance the impact of this paper to a wider audience. It would be particularly interesting if there would not be any stimulation triggered LFP effect and only the transient spiking effect, because it will reveal that the authors found the right modality to reveal causality of phase dependent spiking responses. The results is ideally mentioned explicitly in the manuscript.

We conducted a parallel analysis with the LFP data, following the same procedures as for ESA. Figure 3 has been updated to incorporate the results of the phase-dependent impact of ICMs on LFP amplitude, revealing a striking similarity to the observed effect in ESA: The strongest effect of ICMs on LFP was observed at the same time and for the same γ -phase range as for ESA.

To answer the question about the potentially longer-lasting effects of ICMs on the LFP, we have included a heatmap illustrating the difference between LFPs following ICMs and those from no-ICM trials for a more extended period here. The heatmap below demonstrates that any effect is short-lived and comparable in scale to that observed in ESA. There is no significant deviation of LFPs following ICMs after around 20 ms.

We have integrated the results of γ -phase-dependent changes in LFP amplitude due to ICMs into our Results section (page 10; Fig. 3B and 3C, legend updated), along with corresponding descriptions in lines 183-194 (results section) and lines. However, the manuscript shows the time range from 5 to 25 ms post-ICM to maintain consistency with the ESA plots. Furthermore, we have outlined the analysis procedure in the Methods section.

The heatmap on the left shows the deviation of the LFP in trials following ICMs and the LFP of trials without ICMs (z-scored), but taken from the same times as a function of the γ -excitability phase at 10 ms. The only significant deviation is the negative deflection between 10 to 20 ms between 50° to -50° .

Addition in results section (page 11-12, lines 183-194):

In line with the additional currents expected as a consequence of the additional spikes evoked in V4, there was a significant response in the LFP (Fig. 3B, encircled by a dashed line) during approximately the same time and for similar γ -phases as observed for the spiking response. The strongest deviation of the LFP following ICM pulses occurred at 13.8 ms after the ICM pulse, reflecting a significantly more negative average extracellular potential (12.8 ms – 14.8 ms, mean z-score: -3.95) compared to the LFP from corresponding trials without ICM (mean z-score: -1.32 ± 0.72 SD, $n_{\text{shuf}} = 10,000$; $p < 0.001$, bootstrap hypothesis testing). The strongest negative deflection of the LFP occurred when the V4 population's excitability cycle was at its peak (at 10 ms after the ICM-pulse). While we observed this significant ICM-pulse-evoked, γ -phase-dependent increase in spiking activity and the corresponding LFP, we found no evidence of a substantial effect of ICM on the phase progression of γ -oscillatory activity itself (SFig. 6, SNotes 4).

Discussion section page 14 line 236-238:

....briefly increased the spiking activity within the postsynaptic local populations of V4 neurons, caused corresponding LFP responses, and resulted in longer RTs and a higher probability that monkeys failed to detect the reappearance of the relevant shape.

And updated the methods section accordingly (page 27 lines 518-521):

The procedure to determine γ -phase-dependent effects of ICM pulses on the LFP was identical to the procedure outlined for the ESA, including the statistical analysis. The only distinction lies in the data used: for the LFP, we employed the artifact-removed raw signal without the additional processing steps required to calculate the ESA.

5.) *The success of the manuscript seem to rely on the high sophistication to place the stimuli so that the V2 and V4 sites are jointly stimulated and an additional stimulus is present in the V4 receptive field. But results and illustrations of this important aspect are not included in this manuscript. The reader will benefit to see more explicitly that the V2 sites and V4 sites do show onset responses to the stimulus. Alternatively, the author may want to show cases where phase dependent stimulation effect does not occur because of suboptimal placement of stimuli. Adding more information on this would make the particular strength of the papers approach more apparent to readers.*

To address this point, we updated the current version with additional information and illustrations. We added a panel to Figure 1 to show the onset responses (and sustained responses) of V2 and V4 sites to the stimuli presented in isolation for both animals. With this panel, we provide readers now with an easier understanding of the experimental setup and the complexity of our approach. We also updated the text:

On page 5 line 69-71:

At the same time, we recorded neural activity at V4 sites that retinotopically matched the microstimulated V2 location, as illustrated by the pattern of spiking responses for V4 and V2 sites in Figure 1D.

On page 5 line 78-80:

Two of the four stimuli were located within the RF of the recorded V4 neurons, which responded to each of them when shown individually (Fig. 1D, left panel).

And in the methods section's pages 23-24 lines 435-443:

For the illustration in Figure 1D, a larger Gaussian kernel was used to calculate the neuronal responses at sites in V2 and V4 to each of the two stimuli presented individually within the V4 RF ($\sigma = 20$ ms, 60 ms window size). The resulting ESA of each recording site and condition was then z-scored by subtracting the mean ESA (over time and trials) during the baseline period of the respective trials (starting 100 ms after baseline onset and ending 100 ms before baseline end) and then dividing by the average of the standard deviations calculated for each time bin over trials within the baseline period. The time courses of the neuronal responses averaged across all recording sites for each of the two task conditions were normalized by dividing with the maximal ESA of both task conditions, separately for each animal.

6.) Please also mention explicitly that the microstimulation pulse did not (or did) have effects on the gaze position. One possible scenario to account for the behavioral effect could include a stimulation effect to offset the foveation point (and/or cause a phase reset), disrupting upstream areas. This is ideally mentioned/discussed explicitly in the text, given evidence in the literature that phases or power of visual gamma activity maybe linked to microsaccadic movements.

We have conducted and included an analysis of potentially ICM-dependent changes in gaze positions in the supplementary information (SFig. 1, page 5). For both monkeys, we observed no significant changes in eye positions following ICMs, compared to changes observed in trials without ICMs.

There is a slight tendency for larger changes in gaze positions following ICMs for monkey T. However, this difference did not reach significance and was very limited, with a maximum difference of average gaze positions of only 0.5 arc minutes (0.00833 degrees of visual angle).-

We have incorporated the findings of this control analysis into the results section (**page 8, lines 127:132**):

To examine if such substantial effects of single ICM pulses on RTs might be linked to ICM-induced changes in the direction of gaze, we compared gaze shifts following ICM application with those during periods without ICM-application (see SFig. 1). For both animals, there were neither phase-dependent nor phase independent systematic effects of the ICM pulses on gaze positions, ruling out the possibility that the observed ICM induced effects on RTs are related to ICM evoked alterations of gaze positions

In the discussion section page page15 line263-264:

(3) The lack of ICM-induced changes in gaze position rules out the possibility that ICM-induced eye movements explain the observed behavioural effects.

and we have updated the methods section to include a detailed description of the analysis procedure on **pages 31-32, lines 609-644**:

ICM-induced changes gaze direction

To assess whether the single ICM pulses induced changes in the direction of gaze, including microsaccades or drifts (SFig. 1), which may explain the behavioral effects associated with ICM pulses, we first filtered the recorded analog eye position signals. We applied a low-pass FIR filter in both forward and backward directions (250 Hz pass, 300 Hz stop at 25 dB suppression, pass-band ripples were restricted to ± 0.025 dB). Subsequently, we downsampled the signals to 1000 Hz.

For each trial, we analyzed the time course of the direction of gaze between 100 ms before and 150 ms after ICM pulses. Since eye movements following ICM in V1 have a latency of at least 50 ms if ICM is applied in the upper cortical layers⁷², we used the mean gaze position in the first 50 ms after the ICM pulse as a reference position. To obtain the time course of the deviation of the gaze direction from this reference position, we computed the Euclidean distance between the reference position and the current position of gaze for each time bin following the reference period (50 to 150 ms post-ICM pulse). This calculation performed for the 100 ms preceding the reference period allowed us to assess the expected variability of comparable data in periods not influenced by ICM.

To investigate whether ICM pulses resulted in potential γ -phase dependent changes in the direction of gaze, we averaged the time courses of gaze deviation following the methodology outlined in the methods section: "Phase dependence of missed responses, RTs, LFP, and ESA". This procedure resulted in a matrix containing the median deviation of the gaze direction from the reference direction as a function of time around an ICM pulse and γ -phase for each recording site. These matrices were averaged across recording sites for each individual animal.

Because of the bias that the average deviation of gaze direction from an initial reference direction becomes progressively larger, we generated a surrogate dataset to subtract this bias. To create the surrogate dataset for a recording site, a random time point between 500 and 400 ms before a randomly chosen actual ICM event was selected. The Euclidean distance between the average direction of gaze in the 50 ms reference period preceding this random time point and the gaze direction values in the subsequent 100 ms was calculated. This procedure was repeated as many times as the number of actual ICM pulses from each recording site. The resulting 100 ms periods of gaze direction deviation values were then randomly assigned to an actual γ -phase value for that recording site, and the γ -phase dependence matrix was calculated as described above. For each animal, the resulting matrices were averaged across recording sites. We replicated this procedure 1,000 times, and the mean across these iterations was used to subtract the inherent bias of increasingly larger deviation of gaze direction over time. Additionally, the surrogate dataset was used to assess potential significant changes in eye position induced by the ICMs (see SFig. 1).

7.) The title of the manuscript would benefit from explicitly mentioning that the results are from visual cortex. There are still open questions on how important gamma phases are in other brain systems (with less focal or dense connectivity) and the paper may not want to insinuate the same effects are independent of the brain system.

Following the suggestion, we have changed the title accordingly.

Reviewer #1, #2 and #3

Thank you again for your valuable input and the time you spent reviewing the manuscript, which helped further improve it.

Reviewer #2

In response to my main point #4, the authors wrote that phase values around 9-10 ms could not be taken directly from the signal, because it might already be influenced by an evoked response. I agree with the authors that the phase estimation should exclude parts of the signal that might already be influenced by an evoked response. However, is this accomplished by their current phase estimation procedure? This procedure first removes the electrical artifacts, then applies a broad band-pass filter in the gamma range bidirectionally and finally uses the Hilbert transform to obtain the gamma phases and amplitudes. The filter is a FIR filter, which is by construction acausal, such that the unfiltered signal at time $T+t$ can influence the filtered signal at time T . The value of t depends on the filter specifications, which are given in the methods. When I enter them into the Matlab Filter Designer, I obtain a filter of order 3290, which together with the sampling rate of 25 kHz corresponds to a filter length of 131.6 ms. If this filter is used for a (standard) centered convolution with the data, the value t is half of the filter length, that is $t=65.8$ ms. The main lobes of the filter kernel are contained within 5-10 ms of the center. These values are of relevant size in the context of this study. If the filter is used for a (non-standard) causal convolution, the same applies, because the filter is applied bidirectionally. With bidirectional application, even a IIR filter, which is by construction causal, leads to the described situation. This problem seems to be recognized in the brain-stimulation field, and appropriate modified phase estimation procedures have been developed, see e.g.

Zrenner C, Galevska D, Nieminen JO, Baur D, Stefanou MI, Ziemann U (2020) The shaky ground truth of real-time phase estimation. *NeuroImage* 214:116761.

Indeed, the filter kernel used during phase estimation has a length of 131.6 ms. We were aware of the possible implications of the kernel length on the phase estimation, therefore we tested and evaluated several procedures to estimate the phase of the gamma-band signal just before the possible response in V4, which avoid or minimize effects of response components on this phase estimate. Since the filter required to isolate the γ -band signal should not introduce nonlinear phase shifts, we could not avoid the long, symmetric kernel of a FIR filter. Therefore, we tested phase estimation approaches similar to those proposed by Zrenner et al. (2020). These approaches substitute the actually measured γ -band signal by an estimated, predicted signal. The artificially substituted signal we tested was also generated by an autoregressive model based on the signals before the potentially affected period. However, due to the fast and substantial variation of the γ -oscillations instantaneous frequency, predictions for more than approximately 5 ms lost predictive power fast and became highly unreliable, with large errors in the estimated phases. Such wrongly estimated phases would have unavoidably dissolved the phase-dependent effects we looked for and we therefore decided against these approaches (that work successfully in other frequency bands, which are more stable over longer periods of time).

For these reasons, we used a symmetric FIR filter followed by phase estimation with the Hilbert transform at 5 ms after the ICM-pulse (i.e. 8-9 ms before the LFP's response peak), and calculated the required phase values 4-5 ms later (i.e. between 9 and 10 ms after the ICM-pulse) based on this phase estimate and the

γ -frequency. Well aware that this approach allows for partial overlap between the filter kernel's main lobes with possible responses in the LFP, we checked whether there was any evidence of a relevant effect on phase estimation. We found an essentially flat distribution of the phase values at 5ms after the ICM-pulse, which did not differ from such distributions without ICM. This was incompatible with a substantial bias of phase estimates to a specific value caused by a response in the LFP. The observation was not really surprising given that these responses in the LFP were so small that no substantial ICM-evoked response in the LFP was observed when averaging across all data segments after an ICM-pulse (independent of its phase).

However, we now repeated this investigation by adapting the prediction-based approach of Zrenner et al. to our γ -oscillatory signals. We optimized the autoregressive model by varying the length of the preceding data segments and the model order. Even under the best conditions we could find, the autoregressive model's predictive power was by far too low for estimating future γ -phases with the required reliability.

As shown in the right panel of the figure below (blue distribution), we compared phase estimates of the true and the predicted signals at a time point 95 ms before the ICM pulse—well outside any period that could be affected by ICM-evoked activity. Phase values were obtained using (1) our described procedure—FIR filtering followed by the Hilbert transform—and (2) a 14 ms forward prediction based on a 200 ms signal segment ending 14 ms before the phase estimation time. Although this setup predicted the signal only for 14 ms (roughly one full period of a γ -oscillation) before the time of phase estimation, the resulting distribution of the differences between the two different phase estimates was very broad (median absolute deviation $>75^\circ$ from the true phase value), demonstrating an insufficient accuracy of the model. This approach would distribute responses that actually occur within a comparatively narrow phase range almost randomly across phase space, confirming our earlier observations and precluding using this prediction-based approach for our actual analyses.

The left panel of the figure shows the same comparison at 5 ms after ICM onset. While the distributions of phase difference angles at both time points are highly similar and statistically indistinguishable, possibly indicating that the small responses in the LFP do not substantially affect phase estimation, we believe that the general lack of predictive power—often yielding near-random phase estimates—precludes drawing reliable conclusions from this comparison.

Nevertheless, we agree that the manuscript should explicitly address the potential effects of responses in the LFP on phase estimation.

To directly investigate how the ICM-induced LFP response may influence our phase estimation—and possibly the study's outcome—we analyzed how adding the actual observed response to real signals affected our phase estimates. We used the raw data (1–5000 Hz, 25 kHz sampling rate, with electrical artifacts removed) from the same periods included in the analysis of phase-dependent ICM effects on ESA and the LFP (Fig. 3).

This data was used to:

1. filter all segments using the same filter as described earlier, and estimate the γ -phase based on the Hilbert transform at time $t = -195$ ms. This time point is 200 ms before the actual phase estimation time and thus well outside any period potentially influenced by ICM-related changes in the signal.
2. add the maximal ICM-evoked response observed in the LFP (Fig. 3B, blue curve in the top panel; also represented by the horizontal blue line in the heatmap) to the raw data. Specifically, we defined the evoked response as the period during which the LFP was negative—starting at 9.72 ms and returning to zero at 22.2 ms after the ICM-pulse (as shown in Fig. 3B). This response was then added to each raw data segment at the corresponding time interval following the planned phase estimation point at $t = -195$ ms, i.e., from -190.28 ms to -177.88 ms. Subsequently, the same filtering and phase estimation procedure as described in 1. was performed. The raw data segments with the added maximum responses were filtered as described earlier, and the γ -phase was estimated based on the Hilbert transform at $t = -195$, preceding the onset of the added response by 4.72 ms.

The leftmost panel of the figure below shows the distribution of differences between the phase estimates from the original data and the same data with the added response. Across all segments, the average absolute difference between the two phase estimates was less than 1° (median absolute difference = 0.56°), with 95% of the differences below 1.47° ($n = 8,502$).

If ICM-evoked responses would consistently distort the estimated γ -phase such that the resulting phase values cluster within a limited phase range around the same phase value, we could have falsely identified a preferred phase, simply due to a distortion of the phase estimates by signal components of the evoked response that the filter dragged into the past. However, for such a scenario, the ICM-evoked LFP response would need to disturb the phase estimation strong enough and consistently so that even phases far apart from an illusive preferred phase would be shifted into the narrow phase window of the observed effective phase range. E.g., in the case of the ESA responses, the effective phase range spans about 90° , meaning that if original phases were distributed randomly across all possible phases—including those up to 180° away—they would have to be shifted by up to $\pm 135^\circ$ to reach at least the borders of the effective phase range. The control above shows that even the maximal LFP responses would only allow comparatively small shifts of less than 2° . Consequently, responses that would occur independent of phase would stay almost equally distributed over the entire phase axis, with almost identical average response size across the axis and in stark contrast to the actual pattern of results.

We also tested whether exceptionally large ICM-evoked responses could be capable of phase estimation distortions that allow for the strong redistribution of phase values necessary to explain the pattern phase dependencies observed in the actual data. To do this, we repeated the procedure described above but

increased the added response by a factor of five. Even under this exaggerated condition, the median absolute difference remained small (2.7°), with 95% of the differences below 7.7°. As an extreme scenario, we further tested responses ten times larger than the actual maximum. Even in this case, the average absolute difference between the original γ -phases and those preceding the added responses was only 5.5°, with 95% of the differences below 14.98°.

These results demonstrate that neither ICM-evoked responses of the actually observed size nor those increased to unrealistic levels are capable of producing phase distortions large enough to account for the observed pattern of narrow, phase-specific effects we observed in the ESA and LFP responses as well as in the behavior. Thus, we are confident that a distortion of the γ -phase estimate cannot explain our results due to ICM-evoked responses.

We included this control analysis into the supplementary information of page 14:

Supplementary Figure 9: Effect of ICM-evoked response on phase estimation. To test whether ICM-evoked responses could bias γ -phase estimation, we compared phase estimates from unperturbed data segments (filtered and Hilbert-transformed) with those from the same segments after adding the maximum average ICM-evoked response observed in the V4 LFP (Fig. 3B, top panel). This data with simulated responses was processed identically (see methods section for details). The **Left panel** shows the distribution of the differences between γ -phase estimates in the data with and without simulated response ($n = 8,502$). The median absolute difference was 0.56°, with 95% of γ -phase differences below 1.47°. The **Middle panel** shows the same analysis but with simulated responses five times larger than the actual maximum average ICM-evoked response observed in the V4 LFP. Even under this exaggerated condition, phase distortions remained minor (median absolute difference: 2.7°, 95% < 7.7°). The **Right panel** shows the same as above but with a simulated response ten times the actually observed maximum response. Despite this extreme manipulation, the disturbance of the γ -phase estimation remained very limited (median absolute difference: 5.5°, 95% < 15.0°).

These analyses show that the phase-dependent effects of ICMs on the ESA and the LFP in V4 (Fig. 3) cannot be explained by disturbances of γ -phase estimation. ESA and LFP responses occurred within a narrow effective phase range spanning about 90° around the most effective phase. Inducing such a bias in a set of actually randomly distributed phase values would need to shift phase values substantially and

systematically by up to $\pm 135^\circ$ to change the uniform distribution into an unimodal distribution mainly within the effective phase range. However, even responses ten times stronger than the maximum actually observed only induced an average phase difference of 5.5° , while the actual maximum response caused changes of just 0.56° on average. These phase changes are far too small compared to the large phase shifts that would be necessary to explain the observed strong concentration of responses within the effective phase range with a bias in the phase estimates due to the ICM-evoked responses in the LFP.

We also updated the methods section accordingly (pages 25-26, lines 469-492):

“To test whether the phase estimate at $t = 5$ ms might be substantially influenced by ICM-evoked responses starting approximately 5 ms later (Fig. 3 A/B), we simulated the effect of the observed responses on γ -phase estimates. For this, we used the same raw data segments (25 kHz sampling rate, electrical artifacts removed) already used to analyze the γ -phase dependent effects on ESA and LFP, but during periods preceding the actual time of phase estimation by 200 ms, which precedes any possible ICM-evoked effect on γ -phase. Each segment was required to fulfill the same criterion for γ -power at the time of γ -phase estimation (here 195 ms before ICM) as used in the actual analysis (see Methods: Data selection, criterion 2). The selected data was then processed in two parallel streams.

(1) In the first stream, each segment was filtered bidirectionally using the broadband γ -filter described above, and the γ -phase at $t = -195$ ms was estimated via Hilbert transformation, following the original procedure for phase estimation. (2) In the second stream, the same raw data segments were used to simulate the effect of ICM-evoked LFP responses on γ -phase estimation. For this, we used as a template the largest average ICM-evoked response observed in the LFP (Fig. 3B, top panel, blue curve), defined as the negative deflection occurring between 9.72 ms and 22.12 ms after ICM onset. This response template was added to each data segment at the same relative time with respect to the γ -phase estimation time (i.e., from -190.28 ms to -177.88 ms) as at the actual estimation time 5 ms after the ICM-pulse, thereby preserving the original temporal relationship between response onset and phase estimation. The γ -phase at $t = -195$ ms was then estimated using the same filtering and Hilbert-based method as applied to the original data. To test the influence of stronger responses, the same procedure was repeated using response templates scaled by factors of 5 and 10. The γ -phase values obtained from both processing streams were compared to quantify the potential influence of ICM-evoked LFP responses on γ -phase estimation (SFig. 9).”